# OFF-POLICY EVALUATION FOR RANKING POLICIES UNDER DETERMINISTIC LOGGING POLICIES

**Koichi Tanaka**
Keio University
kouichi_1207@keio.jp

**Kazuki Kawamura**
Sony Group Corporation
Kazuki.Kawamura@sony.com

**Takanori Muroi**
Sony Group Corporation
Takanori.Muroi@sony.com

**Yusuke Narita**
Yale University
yusuke.narita@yale.edu

**Yuki Sasamoto**
Sony Group Corporation
Yuki.Sasamoto@sony.com

**Kei Tateno**
Sony Group Corporation
Kei.Tateno@sony.com

**Takuma Udagawa**
Sony Group Corporation
Takuma.Udagawa@sony.com

**Wei-Wei Du**
Sony Group Corporation
weiwei.du@sony.com

**Yuta Saito**
Hanuku-kaso, Co., Ltd.
saito@hanjuku-kaso.com

## ABSTRACT

Off-Policy Evaluation (OPE) is an important practical problem in algorithmic ranking systems, where the goal is to estimate the expected performance of a new ranking policy using only offline logged data collected under a different, logging policy. Existing estimators, such as the ranking-wise and position-wise inverse propensity score (IPS) estimators, require the data collection policy to be sufficiently stochastic and suffer from severe bias when the logging policy is fully deterministic. In this paper, we propose novel estimators, **Click-based Inverse Propensity Score (CIPS)**, exploiting the intrinsic stochasticity of user click behavior to address this challenge. Unlike existing methods that rely on the stochasticity of the logging policy, our approach uses click probability as a new form of importance weighting, enabling low-bias OPE even under deterministic logging policies where existing methods incur substantial bias. We provide theoretical analyses of the bias and variance properties of the proposed estimators and show, through synthetic and real-world experiments, that our estimators achieve significantly lower bias compared to strong baselines, for a range of experimental settings with completely deterministic logging policies.

## 1 INTRODUCTION

Many intelligent systems, such as recommender systems, search engines, and e-commerce platforms, present items in the form of rankings. These systems interact with users through the *contextual bandit* process, where a ranking policy repeatedly observes a context, produces a ranking, and observes rewards (Kiyohara et al., 2022). As ranking policies interact with the environment, they collect logged data that is valuable for *Off-Policy Evaluation* (OPE). OPE allows researchers to estimate the performance of new policies using only logged data, reducing potential risks and ethical concerns associated with repeated A/B testing (Saito & Joachims, 2021; Dudík et al., 2014; Su et al., 2020; Shimizu et al., 2024).

Table 1: Examples of clicks, potential rewards, and observable rewards.

| position | action | observable reward $CR$ | click $C$ | potential reward $R$ |
|---|---|---|---|---|
| $k = 1$ | $a_1$ | 0 | 0 | 3000 |
| $k = 2$ | $a_2$ | 1000 | 1 | 1000 |
| $k = 3$ | $a_3$ | 0 | 0 | 2000 |
| $k = 4$ | $a_4$ | 0 | 0 | 500 |

A typical OPE study often considers the Direct Method (DM), Inverse Propensity Score (IPS), and Doubly Robust (DR) as the baseline estimators (Saito & Joachims, 2021). DM predicts rewards using a learned model, but it can suffer from substantial bias when the model is misspecified. IPS reweights observed rewards by the ratio of the probabilities of taking the observed action, and DR combines both to reduce variance while being unbiased (Dudík et al., 2014). However, these typical estimators become impractical in ranking problems with extremely large action spaces, because the probability of observing the exact same ranking is often negligible (McInerney et al., 2020).

Although many estimators for the ranking setup already exist to address the challenges of large ranking action spaces (Li et al., 2018; Kiyohara et al., 2022; McInerney et al., 2020), they require the logging policy to be sufficiently stochastic to produce low-bias estimates. For example, the ranking-wise IPS estimator, which reweights observed rewards by the ratio of ranking-wise probabilities, requires the logging policy to explore all the rankings that a new policy may select. The independent IPS estimator, which uses the ratio of position-wise probabilities, requires the logging policy to select all actions at each position (Li et al., 2018). Because of this strong dependence on the stochasticity of the logging policy, these existing estimators can only evaluate the value of actions that the logging policy explored sufficiently (Sachdeva et al., 2020). However, large-scale industrial systems often deploy deterministic policies, as the large action spaces make stochastic policies inefficient and risky to use. In such cases, existing methods suffer from severe bias, making them impractical.

To address the substantial bias caused by deterministic logging policies, we propose a novel class of estimators, namely **Click-based Inverse Propensity Score (CIPS)**, which newly exploits the intrinsic stochasticity of user click behavior. In the ranking setting, a policy recommends a ranked list of multiple actions (e.g., videos, products, or job listings) to an incoming user. Therefore, even if the logging policy is fully deterministic, the user may still view and click each action in the presented ranking with some non-zero probability. Building on this property of ranking settings and user behavior, our approach uses marginalized click probabilities under the new and logging policies to define a new form of importance weights, unlike existing methods that depend solely on the stochasticity of the logging policy. This design enables unbiased or low-bias OPE even in situations where deterministic logging policies cause existing methods to produce substantial bias.

In our theoretical analysis, we show that CIPS achieves unbiasedness under the stochasticity of user click behavior and a relatively mild independence condition on the reward structure. This is, to the best of our knowledge, the first method to provide low-bias value estimation of new ranking policies under completely deterministic logging. Extensive experiments show that our estimator achieves lower mean squared error than existing methods in fully deterministic logging scenarios.

## 2 OFF-POLICY EVALUATION FOR RANKING

This section formulates OPE for ranking policies in the contextual bandit process following existing relevant studies (Li et al., 2018; Kiyohara et al., 2022; 2023; McInerney et al., 2020). Let $x \in \mathcal{X} \subseteq \mathbb{R}^{d_x}$ be a context vector, such as user demographics, sampled from an underlying distribution $p(x)$. We use $\mathcal{A}$ to denote a discrete set of unique actions, with $a \in \mathcal{A}$ corresponding to a specific action such as a movie, news article, or product. Let $A = (a_1, a_2, \cdots, a_{K-1}, a_K)$ be a ranking vector, where $K$ denotes the ranking length. We use $A(k)$ to denote the action at the $k$-th position. The function $\pi : \mathcal{X} \to \Delta(\prod_K(\mathcal{A}))$ is called a *ranking policy*, where $\prod_K(\mathcal{A})$ represents the set of $K$-permutations of $\mathcal{A}$, i.e., the ranking space, and $\Delta(\cdot)$ is the probability simplex.

To closely reflect realistic ranking scenarios, we introduce two types of rewards: *clicks* and *potential rewards*. The latter refers to downstream outcomes (e.g., product purchase, movie play) that may occur after a click, thus modeling the two-stage decision process of users. Let first $C$ denote a

Table 2: A toy example of importance weight under completely deterministic logging policy.

(a) Logging and new policy

|  | $A_1$ | $A_2$ | $A_3$ | $A_4$ | $A_5$ | $A_6$ |
|---|---|---|---|---|---|---|
| $k = 1$ | $a_1$ | $a_1$ | $a_2$ | $a_2$ | $a_3$ | $a_3$ |
| $k = 2$ | $a_2$ | $a_3$ | $a_1$ | $a_3$ | $a_1$ | $a_2$ |
| $k = 3$ | $a_3$ | $a_2$ | $a_3$ | $a_1$ | $a_2$ | $a_1$ |
| $\pi(A|x_1)$ | 0.1 | 0.3 | 0.3 | 0.1 | 0.0 | 0.2 |
| $\pi_0(A|x_1)$ | 1.0 | 0.0 | 0.0 | 0.0 | 0.0 | 0.0 |

(b) Importance weights of IPS

|  | $A_1$ | $A_2$ | $A_3$ | $A_4$ | $A_5$ | $A_6$ |
|---|---|---|---|---|---|---|
| $w(x_1, A)$ | 0.1 | NA | NA | NA | NA | NA |

(c) Importance weights of IIPS

| $w(x_1, A(k))$ | $a_1$ | $a_2$ | $a_3$ |
|---|---|---|---|
| $k = 1$ | 0.4 | NA | NA |
| $k = 2$ | NA | 0.3 | NA |
| $k = 3$ | NA | NA | 0.4 |

click vector sampled from an unknown distribution conditional on the presented ranking, $p(C|x, A)$, where $C(k)$ is a binary signal indicating whether a click occurs at position $k$. Let $R$ then denote a *potential* reward vector drawn from $p(R|x, A)$. We refer to $R$ as "potential" because it is only observable through $CR$, meaning that we observe $R(k)$ only if $C(k) = 1$, i.e., after a click. This structure is based on real-world ranking systems, such as product searches, video recommendations, and many other large-scale e-commerce or content platforms, where user interactions naturally follow two or more sequential phases, i.e., exposure and click, followed by downstream engagement (e.g., purchase, watch time, or dwell time). Table 1 provides an example of clicks and potential rewards. For instance, we observe $R(2)$ because action $a_2$ is clicked at the 2nd position, while we cannot observe $R(1)$, $R(3)$, or $R(4)$ because $a_1$, $a_3$, and $a_4$ are not clicked. Thus, the potential reward vector $R$ is never fully observable on its own.

As a logging policy $\pi_0$ interacts with the environment, we observe $C$ and $CR$ as the rewards corresponding to the selected rankings $A$. Therefore, we obtain the logged dataset

$$\mathcal{D} := \{(x_i, A_i, C_i, C_i R_i)\}_{i=1}^n \sim \prod_{i=1}^n p(x_i)\pi_0(A_i \,|\, x_i)p(C_i R_i \,|\, x_i, A_i),$$

which contains $n$ independent observations drawn under $\pi_0$.

The ultimate goal of OPE is to accurately estimate the effectiveness of a new ranking policy using only the logged data collected under a logging policy that differs from the new policy. We typically define the effectiveness of a ranking policy $\pi$ as (McInerney et al., 2020; Kiyohara et al., 2022)

$$V(\pi) = \mathbb{E}_{p(x)\pi(A|x)p(C,R|x,A)} \left[ \sum_{k=1}^K C(k)R(k) \right] = \mathbb{E}_{p(x)\pi(A|x)} \left[ \sum_{k=1}^K q_k(x, A) \right], \qquad (1)$$

where $q_k(x, A) = \mathbb{E}[C(k)R(k) \mid x, A]$ is the *position-wise* expected reward function regarding position $k$. This quantity $V(\pi)$, referred to as the *policy value*, represents the expected sum of rewards (e.g., sales or play duration) obtained under the deployment of policy $\pi$, and serves as a natural measure of policy performance.

In OPE, we aim to construct an estimator $\hat{V}(\pi; \mathcal{D})$ that can accurately estimate the policy value using only the logged data $\mathcal{D}$. The accuracy of an estimator is typically measured by the mean squared error (MSE) (Uehara et al., 2022):

$$\text{MSE}(\hat{V}) := \mathbb{E}_{p(\mathcal{D})} \left[ \left( \hat{V}(\pi; \mathcal{D}) - V(\pi) \right)^2 \right] = \text{Bias} \left[ \hat{V}(\pi; \mathcal{D}) \right]^2 + \text{Var} \left[ \hat{V}(\pi; \mathcal{D}) \right],$$

where $\mathbb{E}_{p(\mathcal{D})}[\cdot]$ denotes the expectation over the distribution of the log $\mathcal{D}$ and

$$\text{Bias} \left[ \hat{V}(\pi; \mathcal{D}) \right] := \mathbb{E}_{p(\mathcal{D})} \left[ \hat{V}(\pi; \mathcal{D}) \right] - V(\pi), \ \text{Var} \left[ \hat{V}(\pi; \mathcal{D}) \right] := \mathbb{E}_{p(\mathcal{D})} \left[ \left( \hat{V}(\pi; \mathcal{D}) - \mathbb{E}_{p(\mathcal{D})} \left[ \hat{V}(\pi; \mathcal{D}) \right] \right)^2 \right].$$

As thoroughly discussed in Appendix A, most existing studies in ranking OPE focus on mitigating high variance. In contrast, under deterministic logging policies, the bias term in the MSE becomes the dominant issue, making bias reduction the central focus of this study.

Table 3: A toy example of click importance weight under completely deterministic logging policy. The new and logging policy are the same as the previous example given in Table 2.

(a) click probability

| $p_c(x_1, a, A)$ | $A_1$ | $A_2$ | $A_3$ | $A_4$ | $A_5$ | $A_6$ |
|---|---|---|---|---|---|---|
| $k = 1$ | 0.8 | 0.5 | 0.7 | 0.2 | 0.4 | 0.4 |
| $k = 2$ | 0.5 | 0.6 | 0.6 | 0.5 | 0.3 | 0.4 |
| $k = 3$ | 0.2 | 0.1 | 0.5 | 0.4 | 0.2 | 0.1 |

(b) importance weights of CIPS

| | $a_1$ | $a_2$ | $a_3$ |
|---|---|---|---|
| $p_c(x_1, a, \pi)$ | 0.55 | 0.39 | 0.48 |
| $p_c(x_1, a, \pi_0)$ | 0.8 | 0.5 | 0.2 |
| $w(x_1, a, \pi, \pi_0)$ | 0.6875 | 0.78 | 2.4 |

**Existing Estimators.** Here, we review notable existing estimators in the literature of ranking OPE and discuss their limitations under deterministic logging. A more comprehensive overview of related work is available in Appendix A.

First, the ranking-wise IPS estimator is considered the most naive baseline (Kiyohara et al., 2023). IPS reweights observed rewards by the ratio of ranking-wise probabilities under two policies as

$$\hat{V}_{\text{IPS}}(\pi; \mathcal{D}) := \frac{1}{n} \sum_{i=1}^{n} \frac{\pi(A_i|x_i)}{\pi_0(A_i|x_i)} \sum_{k=1}^{K} C_i(k) R_i(k) \quad (2)$$

where $w(x, A) = \pi(A|x)/\pi_0(A|x)$ is called the *ranking-wise importance weight*. This estimator is unbiased under the *ranking-wise common support* condition in the following.

**Condition 2.1** (Ranking-wise Common Support). *The logging policy $\pi_0$ has common support if $\pi(A|x) > 0 \Rightarrow \pi_0(A|x) > 0$ for all $A \in \Delta(\prod_K(\mathcal{A}))$ and $x \in \mathcal{X}$.*

Although ranking-wise IPS only requires Condition 2.1 for an unbiased evaluation, this condition is never satisfied under a deterministic logging policy. To provide intuition for the common support violation, Table 2 presents a toy example with $\mathcal{X} = \{x_1\}$, $\mathcal{A} = \{a_1, a_2, a_3\}$, and $K = 3$. Table 2a shows the ranking probabilities of the new and logging policies. The logging policy selects ranking $A_1$ with probability 1 and never selects any other ranking, meaning it is completely deterministic. Table 2b shows the corresponding ranking-wise importance weights. For example, $w(x_1, A_1) = \pi(A_1|x_1)/\pi_0(A_1|x_1) = 0.1/1.0 = 0.1$, whereas for the other rankings, the weight is undefined due to condition violations. This example illustrates that the common support condition is heavily violated under deterministic logging, leading IPS to incur substantial bias (Sachdeva et al., 2020).

**Theorem 2.1** (Bias of IPS). *IPS produces the following bias under violated Condition 2.1:*

$$\text{Bias}\left(\hat{V}_{\text{IPS}}(\pi; \mathcal{D})\right) = -\mathbb{E}_{p(x)}\left[\sum_{A \in \mathcal{U}_0(x, \pi_0)} \pi(A|x) \sum_{k=1}^{K} q_k(x, A)\right],$$

*where $\mathcal{U}_0(x, \pi_0) = \{A \in \Delta(\prod_K(\mathcal{A})) \mid \pi_0(A|x) = 0\}$ is the set of unsupported rankings for $x$.*

Theorem 2.1 shows that IPS suffers from downward bias, or underestimation, proportional to the number of unsupported rankings, which becomes large under deterministic logging. This occurs because IPS assigns zero reward estimates to unsupported actions. In Table 2b, the set of unsupported rankings is $\mathcal{U}_0 = \{A_2, A_3, A_4, A_5, A_6\}$, meaning almost all available rankings are unsupported. As a result, the bias becomes substantial, posing a critical challenge for accurate OPE.

In the typical literature on ranking OPE with sufficiently stochastic logging policies, the most prominent challenge is that ranking-wise IPS suffers from severe variance due to large action spaces. To mitigate this, other estimators incorporate user behavior conditions (Li et al., 2018; Kiyohara et al., 2022; McInerney et al., 2020). For example, under the *independence condition*, which posits that users interact with items without being influenced by other items in a ranking, the independent IPS (IIPS) estimator can be defined as follows (Li et al., 2018).

$$\hat{V}_{\text{IIPS}}(\pi; \mathcal{D}) := \frac{1}{n} \sum_{i=1}^{n} \sum_{k=1}^{K} \frac{\pi(A_i(k)|x_i)}{\pi_0(A_i(k)|x_i)} C_i(k) R_i(k) \quad (3)$$

where $w(x, A(k)) = \pi(A(k)|x)/\pi_0(A(k)|x)$ is the *position-wise importance weight*, and $\pi(A(k)|x) = \sum_{A'} \pi(A'|x)\mathbb{I}\{A'(k) = A(k)\}$. To ensure unbiased evaluation, IIPS requires the logging policy to satisfy the following exploration condition.

**Condition 2.2** (Position-Wise Common Support). *The logging policy $\pi_0$ has position-wise common support if $\pi(A(k)|x) > 0 \Rightarrow \pi_0(A(k)|x) > 0$ for all $k$, $a \in \mathcal{A}$, and $x \in \mathcal{X}$.*

Although Condition 2.2 imposes weaker stochasticity requirements than Condition 2.1 (needed for unbiasedness of ranking-wise IPS), it is still severely violated under a fully deterministic logging policy. Tables 2b and 2c provide intuition for the distinction between the two versions of common support. In this example, the ranking-wise common support fails in 5 of 6 cases, whereas the position-wise counterpart fails in 6 of 9, indicating that IIPS satisfies its support condition slightly more often than IPS. Yet under deterministic logging, the position-wise common support condition is never satisfied either, which leads IIPS to incur substantial bias.

Other than ranking-wise IPS and IIPS, McInerney et al. (2020) propose RIPS under the *cascade condition*, which posits that users examine items sequentially from the top position downward.

$$\hat{V}_{\text{RIPS}}(\pi; \mathcal{D}) := \frac{1}{n} \sum_{i=1}^{n} \sum_{k=1}^{K} \frac{\pi(A_i(1:k) \mid x_i)}{\pi_0(A_i(1:k) \mid x_i)} C_i(k) R_i(k), \tag{4}$$

where $\pi(A(1:k) \mid x) = \sum_{A'} \pi(A' \mid x)\mathbb{I}\{A'(1:k) = A(1:k)\}$. RIPS requires the logging policy to be stochastic, like IPS and IIPS, and the user behavior to follow the cascade condition for unbiasedness. However, as shown in Appendix B, these requirements are also violated under deterministic logging. To enable effective OPE even in the practical scenario of deterministic logging, we introduce a novel idea designed to circumvent the severe bias caused by the lack of exploration.

## 3 THE PROPOSED ESTIMATOR

This section proposes a new estimator that mitigates the excessive bias caused by fully deterministic logging policies. The key idea is to exploit the intrinsic stochasticity of user click behavior rather than relying on the stochasticity of the logging policy.

To introduce our method, we first focus on user click behavior for each action and reformulate the policy value from Eq. 1 as

$$V(\pi) = \mathbb{E}_{p(x)\pi(A|x)p(C,R|x,A)} \left[ \sum_{a \in \mathcal{A}} C(a)R(a) \right], \tag{5}$$

where $C(a)$ and $R(a)$ denote the click indicator and the potential reward for action $a$, respectively. This reformulation replaces the summation over positions $k$ in Eq. 1 with a summation over actions $a$ and serves merely as a different representation of the same quantity to aid our analysis. It is worth noting that this notation does not impose any independence assumption, i.e., it can generally depend on the actions presented at different positions. For example, the independence assumption used in the IIPS estimator can be written as $\mathbb{E}[C(a)R(a)|A] = \mathbb{E}[C(a)R(a)|a]$, which shows that our notation itself does not imply independence.

We then propose our new estimator, **Click-based Inverse Propensity Score (CIPS)**, which leverages the ratio of click probabilities as a new form of importance weighting.

$$\hat{V}_{\text{CIPS}}(\pi; \mathcal{D}) := \frac{1}{n} \sum_{i=1}^{n} \sum_{a \in \mathcal{A}} \frac{p_c(x_i, a, \pi)}{p_c(x_i, a, \pi_0)} \cdot C_i(a) R_i(a) \tag{6}$$

where $p_c(x, a, \pi)/p_c(x, a, \pi_0)$ is the *click importance weight*, and

$$p_c(x, a, \pi) = \mathbb{E}_{\pi(A|x)}[p_c(x, a, A)] = \mathbb{E}_{\pi(A|x)}[\mathbb{E}[C(a)|x, A]]$$

is the marginalized probability that a user with context $x$ clicks action $a$ under a ranking policy $\pi$.

The key idea behind our method is that, even if the logging policy is fully deterministic, the user may still view each action with a positive probability. We leverage this property by weighting with click-based probabilities.

**Theoretical Analysis.** We first analyze the bias of CIPS under *click-based* common support, which is much less restrictive than the previous support conditions required by existing estimators.

**Condition 3.1** (Click-wise Common Support). *The logging policy $\pi_0$ has click-wise common support if $p_c(x, a, \pi) > 0 \Rightarrow p_c(x, a, \pi_0) > 0$ for all $a \in \mathcal{A}$ and $x \in \mathcal{X}$.*

This condition is more likely to hold because click probabilities are inherently stochastic. Table 3 illustrates the advantage of using click importance weights under a deterministic logging policy. As discussed earlier, existing estimators fail to meet their respective support conditions in this setting by definition. In contrast, Condition 3.1 is satisfied even when the logging policy is deterministic, intuitively showing that CIPS can mitigate bias by leveraging the stochasticity of click behavior.

Additionally, we formally introduce the *independence of potential rewards* condition, which is needed for formally showing when CIPS becomes unbiased.

**Condition 3.2** (Independence of Potential Rewards). *The expected potential rewards satisfy independence if $\mathbb{E}[R(a) \mid x, A] = \mathbb{E}[R(a) \mid x] = q_r(x, a)$ for all $a \in \mathcal{A}$ and $x \in \mathcal{X}$.*

Condition 3.2 states that the expected potential reward for an action depends only on that action $a$, not on other actions in the ranking. This is generally more realistic than the independence condition required by IIPS (Li et al., 2018). For example, in e-commerce it means that, once a user clicks on a product in search, their decision to purchase after a click depends solely on that product's attributes.

Under Conditions 3.1 and 3.2, CIPS is unbiased.

**Theorem 3.1.** *Under Conditions 3.1 and 3.2, CIPS is unbiased, i.e., $\mathbb{E}_{p(\mathcal{D})}[\hat{V}_{\text{CIPS}}(\pi; \mathcal{D})] = V(\pi)$. See Appendix C.1 for the proof.*

Theorem 3.1 shows that CIPS stays unbiased even with deterministic logging, provided these two conditions hold. This is unlike IPS, IIPS, and RIPS, as their stricter support conditions are never satisfied with deterministic logging, leading to their unavoidable and often substantial bias.

It is worth noting that the above bias analysis assumes access to the true click probabilities, which we rarely know in reality. In practice, these probabilities must be estimated from logged data. We can still characterize the bias of CIPS in this case.

**Theorem 3.2.** *Under Conditions 3.1 and 3.2, CIPS has the following bias when using estimated click probabilities $\hat{p}_c(x, a, \pi)$:*

$$\text{Bias}\left(\hat{V}_{\text{CIPS}}\right) = \mathbb{E}_{p(x)}\left[\sum_{a \in \mathcal{A}} p_c(x, a, \pi_0) \left(\frac{\hat{p}_c(x, a, \pi)}{\hat{p}_c(x, a, \pi_0)} - \frac{p_c(x, a, \pi)}{p_c(x, a, \pi_0)}\right) q_r(x, a)\right], \quad (7)$$

*where $\hat{p}_c(x, a, \pi) = \mathbb{E}_{\pi(A|x)}[\hat{p}_c(x, a, A)]$, $w(x, a, \pi, \pi_0) = p_c(x, a, \pi)/p_c(x, a, \pi_0)$, and $q_r(x, a) = \mathbb{E}[R(a) \mid x]$.*

Theorem 3.2 shows that the bias depends on the discrepancy between the ratios of estimated and true click probabilities. Thus, even if the click probabilities themselves are not perfectly accurate, the bias remains small as long as these ratios are estimated well. Empirically, we will show that CIPS performs much better than existing methods by substantially reducing the bias even when the click probabilities are estimated based only on observable data.

Next, we analyze the variance of CIPS.

**Theorem 3.3** (Variance of CIPS). *Under Conditions 3.1 and 3.2, CIPS has the following variance:*

$$n\mathbb{V}_{\mathcal{D}}\left[\hat{V}_{\text{CIPS}}\right] = \sum_{a \in \mathcal{A}} \Big\{ \mathbb{E}_{p(x)\pi_0(A|x)}\left[w^2(x, a, \pi, \pi_0)\sigma^2(x, a, A)\right] \quad (8)$$

$$+ \mathbb{E}_{p(x)}\left[w^2(x, a, \pi, \pi_0)q_r(x, a)^2\mathbb{V}_{\pi_0(A|x)}\left[q_c(x, A)\right]\right] + \mathbb{V}_{p(x)}\left[q_c(x, a, \pi)q_r(x, a)\right] \Big\},$$

*where $\sigma^2(x, a, A) = \mathbb{V}[C(a)R(a) \mid x, A]$.*

Theorem 3.3 shows that the variance of CIPS depends on the magnitude of the click importance weights, which remain stable in deterministic logging. For further variance reduction, it is straightforward to extend CIPS by including weight clipping, self-normalization, or other recent variance reduction techniques. In the following section, seeking a better bias–variance trade-off, we develop the **Click-based Doubly Robust (CDR)** estimator, an extension of CIPS, that lowers variance without introducing additional bias.

# 4 EXTENSION TO CDR

We now extend CIPS to a more sophisticated estimator, **Click-based Doubly Robust (CDR)**, by incorporating a regression model for the expected potential reward.

$$\hat{V}_{\text{CDR}}(\pi; \mathcal{D})$$

$$:= \frac{1}{n} \sum_{i=1}^{n} \sum_{a \in \mathcal{A}} \frac{p_c(x_i, a, \pi)}{p_c(x_i, a, \pi_0)} \cdot (C_i(a)R_i(a) - p_c(x_i, a, A)\hat{q}_r(x_i, a)) + \mathbb{E}_{\pi(A|x_i)} \left[ p_c(x_i, a, A)\hat{q}_r(x_i, a) \right],$$

(9)

where $p_c(x, a, A) = \mathbb{E}[C(a) \mid x, A]$.

We first analyze the bias of CDR under Conditions 3.1 and 3.2, both with and without access to the true click probability.

**Theorem 4.1.** *Under Condition 3.1 and 3.2, CDR is unbiased, i.e., $\mathbb{E}_{p(\mathcal{D})}[\hat{V}_{\text{CDR}}(\pi; \mathcal{D})] = V(\pi)$. See Appendix C.4 for the proof.*

**Theorem 4.2.** *Under Condition 3.1 and 3.2, CDR has the following bias for a given estimated click probability.*

$$\text{Bias}\left(\hat{V}_{\text{CDR}}\right) = \text{Bias}\left(\hat{V}_{\text{CIPS}}\right) = \mathbb{E}_{p(x)} \left[ \sum_a p_c(x, a, \pi_0) \left( \frac{\hat{p}_c(x, a, \pi)}{\hat{p}_c(x, a, \pi_0)} - \frac{p_c(x, a, \pi)}{p_c(x, a, \pi_0)} \right) q_r(x, a) \right]$$

Theorems 4.1 and 4.2 show that CDR produces exactly the same bias as CIPS, regardless of the accuracy of the regression model.

Next, we analyze the variance of CDR and show that it is often smaller than that of CIPS.

**Theorem 4.3** (Variance of CDR). *Under Conditions 3.1 and 3.2, CDR has the following variance.*

$$n\mathbb{V}_{\mathcal{D}}\left[\hat{V}_{\text{CDR}}\right] = \sum_{a \in \mathcal{A}} \left\{ \mathbb{E}_{p(x)\pi_0(A|x)} \left[ w^2(x, a, \pi, \pi_0)\sigma^2(x, a, A) \right] \right.$$

$$\left. + \mathbb{E}_{p(x)} \left[ w^2(x, a, \pi, \pi_0) \cdot \Delta_r^2(x, a)\mathbb{V}_{\pi_0(A|x)}[p_c(x, a, A)] \right] + \mathbb{V}_{p(x)} \left[ p_c(x, a, \pi)q_r(x, a) \right] \right\},$$

*where $\Delta_r(x, a) = q_r(x, a) - \hat{q}_r(x, a)$.*

The key difference between the variance of CIPS (Theorem 3.3) and CDR lies in the term $\Delta_r(x, a)$, which captures the error of the potential reward model. When $\hat{q}_r(x, a)$ is more accurate than simple zero-filling, CDR is expected to reduce variance relative to CIPS.

# 5 EMPIRICAL EVALUATION

In this section, we first evaluate CIPS under deterministic logging policies using synthetic data, aiming to evaluate and compare the proposed methods against the baselines under fully controlled environments. We then assess the real-world applicability of the proposed method on a public recommendation dataset. The implementation code is shared as a supplementary material for reproducibility, and detailed experimental setups can be found in Appendix D.

## 5.1 SYNTHETIC DATA

To generate synthetic data for our experiment, we first randomly sample 10-dimensional context vectors $(x)$ from the standard normal distribution. We then synthesize the expected click probability and the expected potential reward functions as follows.

$$q_c(x, A(k)) = \frac{1}{k} \cdot \text{sigmoid}(\hat{q}_c(x, A(k)) + \sum_{l \neq k} \frac{1}{|k - l|} \cdot \mathbb{W}_c(A(l), A(k))), \tag{10}$$

$$q_r(x, A(k)) = \hat{q}_r(x, A(k)) + \lambda \cdot \sum_{l \neq k} \frac{1}{|k - l|} \cdot \mathbb{W}_r(A(l), A(k)). \tag{11}$$

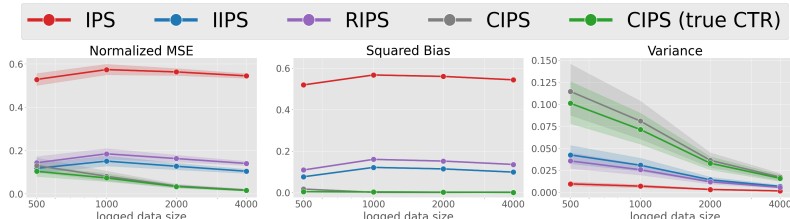

Figure 1: MSE, Squared Bias, and Variance with varying logged data sizes ($n$).

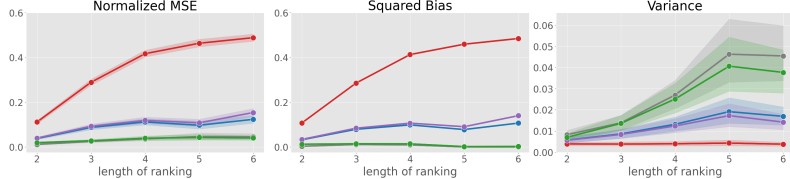

Figure 2: MSE, Squared Bias, and Variance with varying ranking lengths ($K$).

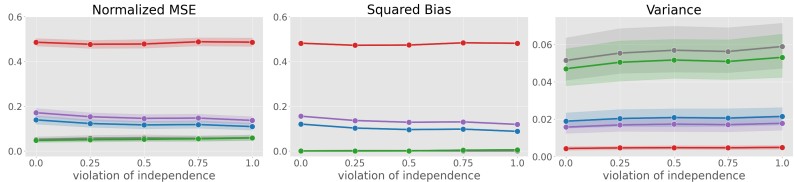

Figure 3: MSE, Squared Bias, and Variance with varying violations of independence ($\lambda$).

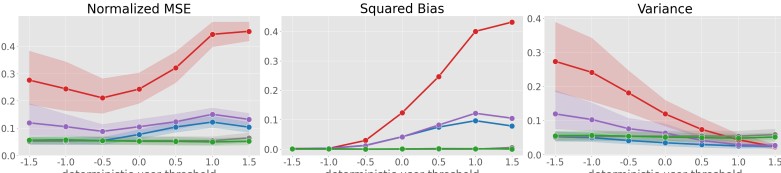

Figure 4: MSE, Squared Bias, and Variance with varying deterministic user thresholds ($\alpha$).

$\mathbb{W}(A(l), A(k))$ denotes the interaction term that measures the effect of action $A(l)$ (presented at position $l$) on the reward of action $A(k)$ (presented at position $k$). We can see that the click probability of $A(k)$ in Eq. 10 depends on both its position $k$ and the entire ranking $A$. We can control the extent of violation of the independence condition through the experimental parameter $\lambda$ in Eq. 11, where the independence condition holds when $\lambda = 0.0$. Based on the above reward functions, we sample the click signal $C(k)$ from a binomial distribution with mean $q_c(x, A(k))$ and the potential reward $R(k)$ from a normal distribution with mean $q_r(x, A(k))$ and standard deviation $\sigma = 1.0$.

We then synthetically define a logging policy $\pi_0$ following the Plackett–Luce model (Plackett, 1975).

$$\pi_0(A|x) = \begin{cases} \prod_{k=1}^{K} \frac{\exp(f_0(x,a(k))),\mathbb{I}[a(k)\notin a_{1:k-1}]}{\sum_{a' \in \mathcal{A} \setminus a_{1:k-1}} \exp(f_0(x,a'))} & \text{if } x^{(1)} > \alpha, \\ \prod_{k=1}^{K} \mathbb{I}\left[a(k) = \underset{a' \in \mathcal{A} \setminus a_{1:k-1}}{\operatorname{argmax}} f_0(x,a')\right] & \text{if } x^{(1)} \leq \alpha \end{cases}, \tag{12}$$

where $f_0(x,a) = \theta_x x + \theta_a a + \theta_{x,a}$. We sample $\theta_x$, $\theta_a$, and $\theta_{x,a}$ from the standard uniform distribution. Here, $x^{(1)}$ denotes the first dimension of the context $x$. The experiment parameter $\alpha$ thus controls the stochasticity of the logging policy; when $x^{(1)} \leq \alpha$, the logging policy for a user with context $x$ becomes completely deterministic. We set $\alpha = \infty$ as the default to evaluate our proposed estimators under a fully deterministic logging policy. In one of the experiments, however, we vary this experimental parameter to compare different methods across a range of determinism levels in the logging policy. We finally define the new policy $\pi$ as

$$\pi(A \mid x) = \prod_{k=1}^{K} \left[ (1-\epsilon) \cdot \mathbb{I}\left[a(k) = \underset{a' \in \mathcal{A} \setminus \{a_{1:k-1}\}}{\operatorname{argmax}} f_1(x,a')\right] + \frac{\epsilon}{|\mathcal{A} \setminus a_{1:k-1}|} \right], \tag{13}$$

where $\epsilon \in [0, 1]$ controls the quality and stochasticity of $\pi$. When $\epsilon = 0.0$, the new policy becomes deterministic and always selects the action that maximizes $f_1(x, a)$ at each position. When $\epsilon = 1.0$, the policy becomes completely uniform. We use $\epsilon = 0.3$ as the default setting.

**Compared methods.** We compare CIPS with IPS, IIPS, and RIPS, whose definitions appear in Section 2. We estimate click probabilities to implement CIPS using a 3-layer neural network and the logged data $\mathcal{D}$. Regarding CIPS, we also report the results of **CIPS (true CTR)**, which uses the true click probability from Eq. 10 just as a reference (CTR refers to Click Through Rate).

### 5.1.1 RESULTS

Figures 1 – 4 present the MSE, squared bias, and variance, each normalized by the true policy value $V(\pi)$. We compute 100 simulations with different random seeds to generate synthetic data and report the averaged results. The shaded regions in the plots represent the 95% confidence intervals estimated with bootstrap. Unless otherwise specified, the logged data size is set to 1000, the number of unique actions is 6, the ranking length is 6, and parameter $\lambda$ in Eq. 11 is 0.5. Note that, beyond the MSE comparison between CIPS and the baselines, we empirically show that CIPS improves policy selection and that applying CDR further reduces estimation variance in Appendix D.2.

**How does CIPS perform when varying the logged data size?** Figure 1 varies the logged data size ($n$) from 500 to 4000. We can see that CIPS consistently outperforms the baselines across all logged data sizes by substantially reducing bias. While the baselines suffer from severe bias due to violations of their support conditions, CIPS achieves low-bias estimates by leveraging the still remaining stochasticity in the click data through its click importance weights. As expected, the variance of the baselines remains small because variance is not a major concern under a completely deterministic logging policy. We also observe that CIPS achieves performance close to that of CIPS (true CTR), although a small additional bias arises from estimating click probabilities $\hat{p}_c(x, a, \pi)$, showing its real-world applicability even without access to the true CTR.

**How does CIPS perform when varying the ranking length?** Figure 2 reports the performance of the estimators as we vary the ranking length $K$. The results show that CIPS provides robust estimations regardless of this experiment parameter. Specifically, CIPS maintains low bias across different lengths, while the bias of the baselines increases with ranking length. Although CIPS exhibits a moderate increase in variance, this effect does not dominate the MSE. Furthermore, its performance remains close to CIPS (true CTR), consistent with the trend we saw in Figure 1.

**How does CIPS perform when violating the independence of potential rewards condition?** In Figure 3, we examine the effect of the independence of potential rewards condition (Condition 3.2) on CIPS, which is one of the necessary conditions for its unbiasedness, by varying $\lambda$ in Eq. 11. Larger values of $\lambda$ correspond to stronger violations of this condition. The results show that the MSE of CIPS increases only marginally as $\lambda$ grows. CIPS continues to provide accurate estimates even when the independence condition is not strictly satisfied and remains the best-performing approach among the compared methods, thanks to its strong bias reduction effect. From these results, we can empirically confirm that the relative advantage of CIPS over existing baselines remains unchanged in a more practical scenario where the independence of potential rewards condition is violated.

**How does CIPS perform with different levels of logging policy stochasticity?** Figure 4 reports the performance of the estimators for varying $\alpha$ in Eq. 12. Larger $\alpha$ values correspond to lower stochasticity in the logging policy. Since the context is sampled from the standard normal distribution, varying $\alpha$ over $[-1.5, \ldots, 1.0, 1.5]$ changes the proportion of users with deterministic logging policies over $[0.07, \ldots, 0.84, 0.93]$. We find that CIPS maintains low bias even at large $\alpha$, in contrast to the baselines, which exhibit substantial bias under near deterministic logging. For smaller $\alpha$ values, where the logging policy is more stochastic and the key bias issues are absent, IIPS performs on par with CIPS. This outcome is expected, as our primary focus is on substantially improving OPE under deterministic environments rather than the typical stochastic setup.

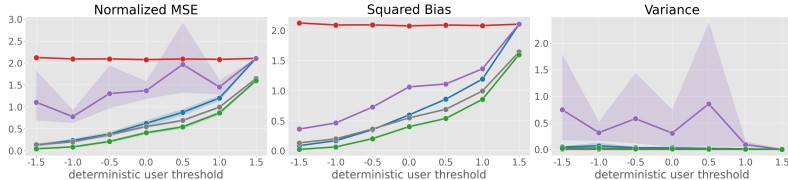

Figure 5: MSE, Squared Bias, and Variance with varying logged data sizes ($n$).

Figure 6: MSE, Squared Bias, and Variance with varying deterministic user thresholds ($\alpha$).

## 5.2 REAL-WORLD DATA

This section conducts ranking OPE experiments on a real-world dataset, KuaiRec (Gao et al., 2022), which contains recommendation logs from the video-sharing application Kuaishou. KuaiRec provides fully observed user–item interactions with nearly 100% density for a subset of users and items. By leveraging this property, we construct OPE experiments with minimal reliance on synthetic components. Specifically, we use the user–item interaction matrix recorded in the original data as the potential reward function $q_r(x, A(k))$ and define the expected click probability as

$$q_c(x, A(k)) = \begin{cases} 1-\eta_{x,A(k)} & \text{if } q_r(x, A(k)) > 2.0 \\ \eta_{x,A(k)} & \text{otherwise} \end{cases}, \tag{14}$$

where $\eta$ is a noise parameter sampled independently from the uniform distribution over $[0, 0.5]$.

We define the logging and new policies following Eq. 12 and Eq. 13 from the synthetic experiments. We conduct real-world experiments with $\epsilon = 0.0$ to simulate the most challenging problem with both deterministic new and logging policies. Note that the number of unique actions is set to 10, the ranking length is 6, and $\alpha = \infty$.

**Results.** Figure 5 compares the estimators under a deterministic new policy while varying the logged data size ($n$). We observe that CIPS consistently achieves lower MSE than the baselines, despite exhibiting some non-negligible bias. This bias stems from violations of the click-wise common support, which can occur when both the new and logging policies are deterministic, an extremely challenging setting for any estimator. Nevertheless, even under partial violations of its common support, CIPS remains more accurate than the baselines, which completely fail under deterministic logging due to full violations of their support conditions.

Figure 6 evaluates the estimators as the deterministic user threshold ($\alpha$) varies. As the logging policy becomes more deterministic, the relative advantage of CIPS over the baselines increases. Although all estimators experience bias growth with increasing $\alpha$, CIPS limits this growth most effectively.

## 6 CONCLUSION

We studied Off-Policy Evaluation (OPE) for ranking under deterministic logging, where existing methods relying on logging-policy stochasticity incur severe bias. To address this previously untackled problem, we proposed the Click-based Inverse Propensity Score estimator, which exploits the intrinsic stochasticity of user click behavior via click importance weights, substantially relaxing the support condition and enabling unbiased or low-bias estimation under deterministic logging. Overall, this work challenges the long-held belief that accurate OPE is infeasible under deterministic logging. Our method makes reliable evaluation possible in this previously intractable setting via the new weighting design, paving the way for broader applicability of OPE in real-world systems.

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

## A    RELATED WORK

In this section, we distinguish our work from related studies and clarify the contributions of our methods and analysis.

First, we review OPE for ranking policies Li et al. (2018); Kiyohara et al. (2022); McInerney et al. (2020); Kiyohara et al. (2023) under the standard setting with sufficiently stochastic logging policies. Prior work mainly addresses the severe variance caused by large ranking spaces. To reduce the high variance of the "naïve" ranking-wise IPS estimator, recent methods introduce user behavior assumptions such as independence Li et al. (2018) or cascade models McInerney et al. (2020); Kiyohara et al. (2022) to improve the bias–variance trade-off. In contrast, we target the severe bias introduced by deterministic logging policies rather than the variance problem. In this setting, all existing estimators fail drastically because they rely on the logging policy's stochasticity and suffer from substantial bias due to the lack of exploration in the logged data.

Second, we compare our contributions with studies addressing the deficient support problem in the typical (non-ranking) OPE formulation Sachdeva et al. (2020); Felicioni et al. (2022). Deficient support refers to a milder situation where the logging policy assigns zero probability to some actions that the new policy may select. This situation is problematic for OPE because importance weighting cannot evaluate the reward of actions that have no probability of being selected under the logging policy. Compared to the deficient support problem, our work addresses an even more challenging setting, a fully deterministic logging policy. Deterministic logging is a strict special case of deficient support, eliminating all stochasticity, which makes existing methods severely ineffective and motivates our development of a new approach tailored to this scenario.

Third, we distinguish our contributions from studies on OPE for large action spaces Saito et al. (2023); Saito & Joachims (2022); Taufiq et al. (2023); Guzman-Olivares et al. (2025); Kiyohara et al. (2024). These studies aim to deal with the severe variance problem caused by large action spaces, similar to the typical OPE studies for ranking Li et al. (2018); McInerney et al. (2020); Kiyohara et al. (2022; 2023). Their key technique is to marginalize importance weights leveraging observed action embeddings to prevent the weights from taking excessively large values. Our proposed estimators also marginalize importance weights, but we do so using click probabilities rather than embeddings. While this shares the general idea, our goal is to address the lack of stochasticity in deterministic logging policies rather than variance in large action spaces. Furthermore, methods for large action spaces require additional information, such as pre-observed action embeddings, which our method does not require.

Finally, we mention a recent work addressing OPE in multiple domains Natsubori et al. (2025). That study uses logged data collected from multiple domains (e.g., multiple countries or hospitals) to compensate for the lack of stochasticity in the logging policy of the target domain. Although it tackles a similar problem to ours, it depends on multi-domain logged data, which is rarely available. Our method addresses deterministic logging without requiring data from multiple domains by exploiting the intrinsic stochasticity of user click behavior.

## B    EXAMPLES OF IMPORTANCE WEIGHTS UNDER COMPLETELY DETERMINISTIC LOGGING POLICY

Here, we provide useful intuition for the common support violation. Table 4 presents a toy example with $\mathcal{X} = \{x_1\}$, $\mathcal{A} = \{a_1, a_2, a_3\}$, and $K = 3$. This example is the same as Table 2 in the main text. In Table 4, we also consider importance weights of RIPS as well as those of ranking-wise IPS and position-wise IPS. As we discussed in the main text, the ranking-wise common support fails in 5 of 6 cases, whereas the position-wise counterpart fails in 6 of 9, indicating that IIPS satisfies its support condition slightly more often than IPS. Table 4d shows importance weights of RIPS. We can see that the common support of RIPS fails in 3 of 15 cases, suggesting that RIPS has severe bias under deterministic logging policies. This also indicates that RIPS satisfies its support condition slightly more often than IPS, but less often than IIPS. We can consider RIPS as an estimator that lies between IPS and IIPS. Thus, all baseline estimators in ranking OPE suffer from violation of support conditions, resulting in introducing substantial bias. However, CIPS achieves low-bias OPE under deterministic logging policies by leveraging click importance weights.

Table 4: A toy example of importance weight under completely deterministic logging policy.

(a) Logging and new policy

|  | $A_1$ | $A_2$ | $A_3$ | $A_4$ | $A_5$ | $A_6$ |
|---|---|---|---|---|---|---|
| $k=1$ | $a_1$ | $a_1$ | $a_2$ | $a_2$ | $a_3$ | $a_3$ |
| $k=2$ | $a_2$ | $a_3$ | $a_1$ | $a_3$ | $a_1$ | $a_2$ |
| $k=3$ | $a_3$ | $a_2$ | $a_3$ | $a_1$ | $a_2$ | $a_1$ |
| $\pi(A\|x_1)$ | 0.1 | 0.3 | 0.3 | 0.1 | 0.0 | 0.2 |
| $\pi_0(A\|x_1)$ | 1.0 | 0.0 | 0.0 | 0.0 | 0.0 | 0.0 |

(b) Importance weights of IPS

|  | $A_1$ | $A_2$ | $A_3$ | $A_4$ | $A_5$ | $A_6$ |
|---|---|---|---|---|---|---|
| $w(x_1, A)$ | 0.1 | NA | NA | NA | NA | NA |

(c) Importance weights of IIPS

| $w(x_1, A(k))$ | $a_1$ | $a_2$ | $a_3$ |
|---|---|---|---|
| $k=1$ | 0.4 | NA | NA |
| $k=2$ | NA | 0.3 | NA |
| $k=3$ | NA | NA | 0.4 |

(d) Importance weights of RIPS

| $A(1:1)$ | $a_1$ | $a_2$ | $a_3$ |
|---|---|---|---|
| $w(x_1, A(1:1))$ | 0.4 | NA | NA |

| $A(1:2)$ | $(a_1, a_2)$ | $(a_1, a_3)$ | $(a_2, a_1)$ | $(a_2, a_3)$ | $(a_3, a_1)$ | $(a_3, a_2)$ |
|---|---|---|---|---|---|---|
| $w(x_1, A(1:2))$ | 0.1 | NA | NA | NA | NA | NA |

| $A(1:3)$ | $(a_1, a_2, a_3)$ | $(a_1, a_3, a_2)$ | $(a_2, a_1, a_3)$ | $(a_2, a_3, a_1)$ | $(a_3, a_1, a_2)$ | $(a_3, a_2, a_1)$ |
|---|---|---|---|---|---|---|
| $w(x_1, A(1:3))$ | 0.1 | NA | NA | NA | NA | NA |

## C  OMITTED PROOFS

Here, we provide the derivations and proofs that are omitted in the main text.

### C.1  PROOF OF THEOREM 3.1

We show that CIPS is unbiased under Conditions 3.1 and 3.2.

$$\mathbb{E}_{p(x)\pi_0(A|x)p(C,R|x,A)}\left[\frac{1}{n}\sum_{i=1}^{n}\sum_{a\in\mathcal{A}}\frac{p_c(x_i,a_i,\pi)}{p_c(x_i,a_i,\pi_0)}C_i(a)R_i(a)\right]$$

$$=\mathbb{E}_{p(x)\pi_0(A|x)}\left[\sum_{a\in\mathcal{A}}\frac{p_c(x,a,\pi)}{p_c(x,a,\pi_0)}\mathbb{E}\left[C(a)R(a)|x,A\right]\right]$$

$$=\mathbb{E}_{p(x)\pi_0(A|x)}\left[\sum_{a\in\mathcal{A}}\frac{p_c(x,a,\pi)}{p_c(x,a,\pi_0)}\mathbb{E}\left[C(a)|x,A\right]\mathbb{E}\left[R(a)|x\right]\right]$$

$$=\mathbb{E}_{p(x)}\left[\sum_{a\in\mathcal{A}}\frac{p_c(x,a,\pi)}{p_c(x,a,\pi_0)}\mathbb{E}\left[R(a)|x\right]\sum_{A}\pi_0(A|x)\mathbb{E}\left[C(a)|x,A\right]\right]$$

$$=\mathbb{E}_{p(x)}\left[\sum_{a\in\mathcal{A}}\frac{p_c(x,a,\pi)}{p_c(x,a,\pi_0)}\mathbb{E}\left[R(a)|x\right]p_c(x,a,\pi_0)\right]$$

$$=\mathbb{E}_{p(x)}\left[\sum_{a\in\mathcal{A}}p_c(x,a,\pi)\mathbb{E}\left[R(a)|x\right]\right]$$

$$=\mathbb{E}_{p(x)}\left[\sum_{a\in\mathcal{A}}\sum_{A}\pi(A|x)\mathbb{E}[C(a)|x,A]\mathbb{E}\left[R(a)|x\right]\right]$$

$$= \mathbb{E}_{p(x)\pi(A|x)p(C,R|x,A)} \left[ \sum_{a\in\mathcal{A}} C(a)R(a) \right]$$

$$= V(\pi)$$

## C.2   PROOF OF THEOREM 3.2

$$\text{Bias}\left(\hat{V}_{\text{CIPS}}\right) = \mathbb{E}_{p(x)\pi_0(A|x)p(C,R|x,A)} \left[ \frac{1}{n}\sum_{i=1}^{n}\sum_{a\in\mathcal{A}} \frac{\hat{p}_c(x_i, a_i, \pi)}{\hat{p}_c(x_i, a_i, \pi_0)} C_i(a)R_i(a) \right] - V(\pi)$$

$$= \mathbb{E}_{p(x)\pi_0(A|x)p(C,R|x,A)} \left[ \sum_{a\in\mathcal{A}} \frac{\hat{p}_c(x, a, \pi)}{\hat{p}_c(x, a, \pi_0)} C(a)R(a) \right] - \mathbb{E}_{p(x)\pi(A|x)p(C,R|x,A)} \left[ \sum_{a\in\mathcal{A}} C(a)R(a) \right]$$

$$= \mathbb{E}_{p(x)\pi_0(A|x)} \left[ \sum_{a\in\mathcal{A}} \frac{\hat{p}_c(x, a, \pi)}{\hat{p}_c(x, a, \pi_0)} \mathbb{E}[C(a)|x, A]\mathbb{E}[R(a)|x, A] \right]$$

$$- \mathbb{E}_{p(x)\pi(A|x)} \left[ \sum_{a\in\mathcal{A}} \mathbb{E}[C(a)|x, A]\mathbb{E}[R(a)|x, A] \right]$$

$$= \mathbb{E}_{p(x)\pi_0(A|x)} \left[ \sum_{a\in\mathcal{A}} \frac{\hat{p}_c(x, a, \pi)}{\hat{p}_c(x, a, \pi_0)} \mathbb{E}[C(a)|x, A]\mathbb{E}[R(a)|x] \right]$$

$$- \mathbb{E}_{p(x)\pi(A|x)} \left[ \sum_{a\in\mathcal{A}} \mathbb{E}[C(a)|x, A]\mathbb{E}[R(a)|x] \right]$$

$$\tag{15}$$

$$= \mathbb{E}_{p(x)} \left[ \sum_{a\in\mathcal{A}} \frac{\hat{p}_c(x, a, \pi)}{\hat{p}_c(x, a, \pi_0)} q_r(x, a) \sum_A \pi_0(A|x)\mathbb{E}[C(a)|x, A] \right]$$

$$- \mathbb{E}_{p(x)} \left[ \sum_{a\in\mathcal{A}} q_r(x, a) \sum_A \pi(A|x)\mathbb{E}[C(a)|x, A] \right]$$

$$= \mathbb{E}_{p(x)} \left[ \sum_{a\in\mathcal{A}} \frac{\hat{p}_c(x, a, \pi)}{\hat{p}_c(x, a, \pi_0)} q_r(x, a)p_c(x, a, \pi_0) \right] - \mathbb{E}_{p(x)} \left[ \sum_{a\in\mathcal{A}} q_r(x, a)p_c(x, a, \pi) \right]$$

$$= \mathbb{E}_{p(x)} \left[ \sum_{a\in\mathcal{A}} p_c(x, a, \pi_0) \left( \frac{\hat{p}_c(x, a, \pi)}{\hat{p}_c(x, a, \pi_0)} - \frac{p_c(x, a, \pi)}{p_c(x, a, \pi_0)} \right) q_r(x, a) \right],$$

where we use Condition 3.2 in Eq. 15.

## C.3   PROOF OF THEOREM 3.3

We can derive the variance of CIPS by setting $\hat{q}_r(x, a) = 0$ in the variance of CDR.

$$n\mathbb{V}_{\mathcal{D}}\left[\hat{V}_{\text{CIPS}}\right] = \sum_{a\in\mathcal{A}} \left\{ \mathbb{E}_{p(x)\pi_0(A|x)} \left[ w^2(x, a, \pi, \pi_0)\sigma^2(x, a, A) \right] \right.$$

$$\left. + \mathbb{E}_{p(x)} \left[ w^2(x, a, \pi, \pi_0)q_r(x, a)\mathbb{V}_{\pi_0(A|x)}\left[ q_c(x, a, A) \right] \right] + \mathbb{V}_{p(x)}\left[ p_c(x, a, \pi)q_r(x, a) \right] \right\},$$

where $\sigma^2(x, a, A) = \mathbb{V}[C(a)R(a) \mid x, A]$.

## C.4   PROOF OF THEOREM 4.1

We show that CDR is unbiased under Conditions 3.1 and 3.2.

$$\mathbb{E}_{p(x)\pi_0(A|x)p(C,R|x,A)} \left[ \frac{1}{n}\sum_{i=1}^{n}\sum_{a\in\mathcal{A}} \frac{p_c(x_i, a, \pi)}{p_c(x_i, a, \pi_0)} \cdot (C_i(a)R_i(a) - p_c(x_i, a, A)\hat{q}_r(x_i, a)) + \mathbb{E}_{\pi(A|x_i)}\left[ p_c(x_i, a, A)\hat{q}_r(x_i, a) \right] \right]$$

$$= \mathbb{E}_{p(x)\pi_0(A|x)p(C,R|x,A)} \left[ \sum_{a \in \mathcal{A}} \frac{p_c(x,a,\pi)}{p_c(x,a,\pi_0)} \cdot (C(a)R(a) - p_c(x,a,A)\hat{q}_r(x,a)) + \mathbb{E}_{\pi(A|x)} [p_c(x,a,A)\hat{q}_r(x,a)] \right]$$

$$= \mathbb{E}_{p(x)\pi_0(A|x)} \left[ \sum_{a \in \mathcal{A}} \frac{p_c(x,a,\pi)}{p_c(x,a,\pi_0)} \cdot (\mathbb{E}[C(a)R(a)|x,A] - p_c(x,a,A)\hat{q}_r(x,a)) + \mathbb{E}_{\pi(A|x)} [p_c(x,a,A)\hat{q}_r(x,a)] \right]$$

$$= \mathbb{E}_{p(x)\pi_0(A|x)} \left[ \sum_{a \in \mathcal{A}} \frac{p_c(x,a,\pi)}{p_c(x,a,\pi_0)} \cdot (p_c(x,a,A)q_r(x,a) - p_c(x,a,A)\hat{q}_r(x,a)) + \mathbb{E}_{\pi(A|x)} [p_c(x,a,A)\hat{q}_r(x,a)] \right]$$

$$= \mathbb{E}_{p(x)} \left[ \sum_{a \in \mathcal{A}} \frac{p_c(x,a,\pi)}{p_c(x,a,\pi_0)} \cdot (q_r(x,a) - \hat{q}_r(x,a)) \sum_A \pi_0(A|x)p_c(x,a,A) + \mathbb{E}_{\pi(A|x)} [p_c(x,a,A)\hat{q}_r(x,a)] \right]$$

$$= \mathbb{E}_{p(x)} \left[ \sum_{a \in \mathcal{A}} \frac{p_c(x,a,\pi)}{p_c(x,a,\pi_0)} \cdot (q_r(x,a) - \hat{q}_r(x,a)) \, p_c(x,a,\pi_0) + p_c(x,a,\pi)\hat{q}_r(x,a) \right]$$

$$= \mathbb{E}_{p(x)} \left[ \sum_{a \in \mathcal{A}} p_c(x,a,\pi) \cdot (q_r(x,a) - \hat{q}_r(x,a)) + p_c(x,a,\pi)\hat{q}_r(x,a) \right]$$

$$= \mathbb{E}_{p(x)} \left[ \sum_{a \in \mathcal{A}} p_c(x,a,\pi)q_r(x,a) \right]$$

$$= \mathbb{E}_{p(x)} \left[ \sum_{a \in \mathcal{A}} \sum_A \pi(A|x)p_c(x,a,A)q_r(x,a) \right]$$

$$= \mathbb{E}_{p(x)\pi(A|x)} \left[ \sum_{a \in \mathcal{A}} p_c(x,a,A)q_r(x,a) \right]$$

$$= \mathbb{E}_{p(x)\pi(A|x)p(C,R|x,A)} \left[ \sum_{a \in \mathcal{A}} C(a)R(a) \right]$$

$$= V(\pi)$$

## C.5 Proof of Theorem 4.2

$$\text{Bias}\left(\hat{V}_{\text{CDR}}\right)$$

$$= \mathbb{E}_{p(\mathcal{D})} \left[ \frac{1}{n} \sum_{i=1}^n \sum_{a \in \mathcal{A}} \frac{\hat{p}_c(x_i,a,\pi)}{\hat{p}_c(x_i,a,\pi_0)} \cdot (C_i(a)R_i(a) - \hat{p}_c(x_i,a,A)\hat{q}_r(x_i,a)) + \mathbb{E}_{\pi(A|x_i)} [\hat{p}_c(x_i,a,A)\hat{q}_r(x_i,a)] \right] - V(\pi)$$

$$= \mathbb{E}_{p(\mathcal{D})} \left[ \sum_{a \in \mathcal{A}} \frac{\hat{p}_c(x,a,\pi)}{\hat{p}_c(x,a,\pi_0)} \cdot (C(a)R(a) - \hat{p}_c(x,a,A)\hat{q}_r(x,a)) + \mathbb{E}_{\pi(A|x)} [\hat{p}_c(x,a,A)\hat{q}_r(x,a)] \right] - V(\pi)$$

$$= \mathbb{E}_{p(x)\pi_0(A|x)} \left[ \sum_{a \in \mathcal{A}} \frac{\hat{p}_c(x,a,\pi)}{\hat{p}_c(x,a,\pi_0)} \cdot (p_c(x,a,A)q_r(x,A) - \hat{p}_c(x,a,A)\hat{q}_r(x,a)) + \hat{p}_c(x,a,\pi)\hat{q}_r(x,a) \right] - V(\pi)$$

$$= \mathbb{E}_{p(x)} \left[ \sum_{a \in \mathcal{A}} \frac{\hat{p}_c(x,a,\pi)}{\hat{p}_c(x,a,\pi_0)} \cdot (p_c(x,a,\pi_0)q_r(x,A) - \hat{p}_c(x,a,\pi_0)\hat{q}_r(x,a)) + \hat{p}_c(x,a,\pi)\hat{q}_r(x,a) \right] - V(\pi)$$

$$= \mathbb{E}_{p(x)} \left[ \sum_{a \in \mathcal{A}} \frac{\hat{p}_c(x,a,\pi)}{\hat{p}_c(x,a,\pi_0)} p_c(x,a,\pi_0)q_r(x,A) \right] - \mathbb{E}_{p(x)} [p_c(x,a,\pi)q_r(x,a)]$$

$$= \mathbb{E}_{p(x)} \left[ \sum_{a \in \mathcal{A}} p_c(x,a,\pi_0) \left( \frac{\hat{p}_c(x,a,\pi)}{\hat{p}_c(x,a,\pi_0)} - \frac{p_c(x,a,\pi)}{p_c(x,a,\pi_0)} \right) q_r(x,a) \right]$$

$$= \text{Bias}\left(\hat{V}_{\text{CIPS}}\right) \tag{16}$$

## C.6 PROOF OF THEOREM 4.3

$$
n\mathbb{V}_{\mathcal{D}}\left[\hat{V}_{\text{CDR}}\right]
$$

$$
= n\mathbb{V}_{\mathcal{D}}\left[\frac{1}{n}\sum_{i=1}^{n}\sum_{a\in\mathcal{A}}\frac{p_c(x_i,a,\pi)}{p_c(x_i,a,\pi_0)}\cdot(C_i(a)R_i(a)-p_c(x_i,a,A)\hat{q}_r(x_i,a))+\mathbb{E}_{\pi(A|x_i)}\left[p_c(x_i,a,A)\hat{q}_r(x_i,a)\right]\right]
$$

$$
= \mathbb{V}_{\mathcal{D}}\left[\sum_{a\in\mathcal{A}}\frac{p_c(x,a,\pi)}{p_c(x,a,\pi_0)}\cdot(C(a)R(a)-p_c(x,a,A)\hat{q}_r(x,a))+\mathbb{E}_{\pi(A|x)}\left[p_c(x,a,A)\hat{q}_r(x,a)\right]\right]
$$

$$
= \mathbb{E}_{p(x)\pi_0(A|x)}\left[\mathbb{V}_{p(C,R|x,A)}\left[\sum_{a\in\mathcal{A}}w(x,a,\pi,\pi_0)\cdot(C(a)R(a)-p_c(x,a,A)\hat{q}_r(x,a))+\mathbb{E}_{\pi(A|x)}\left[p_c(x,a,A)\hat{q}_r(x,a)\right]\right]\right]
$$

$$
\quad + \mathbb{V}_{p(x)\pi_0(A|x)}\left[\mathbb{E}_{p(C,R|x,A)}\left[\sum_{a\in\mathcal{A}}w(x,a,\pi,\pi_0)\cdot(C(a)R(a)-p_c(x,a,A)\hat{q}_r(x,a))+\mathbb{E}_{\pi(A|x)}\left[p_c(x,a,A)\hat{q}_r(x,a)\right]\right]\right]
$$

$$
= \mathbb{E}_{p(x)\pi_0(A|x)}\left[\sum_{a\in\mathcal{A}}w^2(x,a,\pi,\pi_0)\sigma^2(x,a,A)\right]
$$

$$
\quad + \mathbb{V}_{p(x)\pi_0(A|x)}\left[\sum_{a\in\mathcal{A}}w(x,a,\pi,\pi_0)\cdot(p_c(x,a,A)q_r(x,a)-p_c(x,a,A)\hat{q}_r(x,a))+\mathbb{E}_{\pi(A|x)}\left[p_c(x,a,A)\hat{q}_r(x,a)\right]\right]
$$

$$
= \mathbb{E}_{p(x)\pi_0(A|x)}\left[\sum_{a\in\mathcal{A}}w^2(x,a,\pi,\pi_0)\sigma^2(x,a,A)\right]
$$

$$
\quad + \mathbb{E}_{p(x)}\left[\mathbb{V}_{\pi_0(A|x)}\left[\sum_{a\in\mathcal{A}}w(x,a,\pi,\pi_0)\cdot(p_c(x,a,A)q_r(x,a)-p_c(x,a,A)\hat{q}_r(x,a))+\mathbb{E}_{\pi(A|x)}\left[p_c(x,a,A)\hat{q}_r(x,a)\right]\right]\right]
$$

$$
\quad + \mathbb{V}_{p(x)}\left[\mathbb{E}_{\pi_0(A|x)}\left[\sum_{a\in\mathcal{A}}w(x,a,\pi,\pi_0)\cdot(p_c(x,a,A)q_r(x,a)-p_c(x,a,A)\hat{q}_r(x,a))+\mathbb{E}_{\pi(A|x)}\left[p_c(x,a,A)\hat{q}_r(x,a)\right]\right]\right]
$$

$$
= \mathbb{E}_{p(x)\pi_0(A|x)}\left[\sum_{a\in\mathcal{A}}w^2(x,a,\pi,\pi_0)\sigma^2(x,a,A)\right]
$$

$$
\quad + \mathbb{E}_{p(x)}\left[\sum_{a\in\mathcal{A}}w^2(x,a,\pi,\pi_0)\cdot(q_r(x,a)-\hat{q}_r(x,a))^2\,\mathbb{V}_{\pi_0(A|x)}[p_c(x,a,A)]\right]
$$

$$
\quad + \mathbb{V}_{p(x)}\left[\sum_{a\in\mathcal{A}}w(x,a,\pi,\pi_0)\cdot(q_r(x,a)-\hat{q}_r(x,a))\sum_{A}\pi_0(A|x)p_c(x,a,A)+\mathbb{E}_{\pi(A|x)}\left[p_c(x,a,A)\hat{q}_r(x,a)\right]\right]
$$

$$
= \mathbb{E}_{p(x)\pi_0(A|x)}\left[\sum_{a\in\mathcal{A}}w^2(x,a,\pi,\pi_0)\sigma^2(x,a,A)\right]
$$

$$
\quad + \mathbb{E}_{p(x)}\left[\sum_{a\in\mathcal{A}}w^2(x,a,\pi,\pi_0)\cdot(q_r(x,a)-\hat{q}_r(x,a))^2\,\mathbb{V}_{\pi_0(A|x)}[p_c(x,a,A)]\right]
$$

$$
\quad + \mathbb{V}_{p(x)}\left[\sum_{a\in\mathcal{A}}w(x,a,\pi,\pi_0)\cdot(q_r(x,a)-\hat{q}_r(x,a))\,p_c(x,a,\pi_0)+\mathbb{E}_{\pi(A|x)}\left[p_c(x,a,A)\hat{q}_r(x,a)\right]\right]
$$

$$
= \mathbb{E}_{p(x)\pi_0(A|x)}\left[\sum_{a\in\mathcal{A}}w^2(x,a,\pi,\pi_0)\sigma^2(x,a,A)\right]
$$

$$
\quad + \mathbb{E}_{p(x)}\left[\sum_{a\in\mathcal{A}}w^2(x,a,\pi,\pi_0)\cdot(q_r(x,a)-\hat{q}_r(x,a))^2\,\mathbb{V}_{\pi_0(A|x)}[p_c(x,a,A)]\right]
$$

$$
\quad + \mathbb{V}_{p(x)}\left[\sum_{a\in\mathcal{A}}p_c(x,a,\pi)\cdot(q_r(x,a)-\hat{q}_r(x,a))+p_c(x,a,\pi)\hat{q}_r(x,a)\right]
$$

$$
= \mathbb{E}_{p(x)\pi_0(A|x)}\left[\sum_{a\in\mathcal{A}}w^2(x,a,\pi,\pi_0)\sigma^2(x,a,A)\right]
$$

$$
\quad + \mathbb{E}_{p(x)}\left[\sum_{a\in\mathcal{A}}w^2(x,a,\pi,\pi_0)\cdot(q_r(x,a)-\hat{q}_r(x,a))^2\,\mathbb{V}_{\pi_0(A|x)}[p_c(x,a,A)]\right]
$$

$$+ \mathbb{V}_{p(x)}\Big[ \sum_{a \in \mathcal{A}} p_c(x, a, \pi) q_r(x, a) \Big]$$

$$= \sum_{a \in \mathcal{A}} \Big\{ \mathbb{E}_{p(x)\pi_0(A|x)}\Big[ w^2(x, a, \pi, \pi_0)\sigma^2(x, a, A) \Big]$$

$$+ \mathbb{E}_{p(x)}\Big[ w^2(x, a, \pi, \pi_0) \cdot \Delta_r^2(x, a) \mathbb{V}_{\pi_0(A|x)}[p_c(x, a, A)] \Big] + \mathbb{V}_{p(x)}\Big[ p_c(x, a, \pi) q_r(x, a) \Big] \Big\}$$

where $\Delta_r(x, a) = q_r(x, a) - \hat{q}_r(x, a)$.

# D  ADDITIONAL EXPERIMENTAL SETUPS AND RESULTS

This section describes the detailed experimental settings and reports additional results.

## D.1  SYNTHETIC EXPERIMENTS

**Detailed Setup.**  We first describe the synthetic experiment settings in detail, followed by how we define the expected reward function. In the synthetic experiments, the expected reward function is defined as follows:

$$q_c(x, A(k)) = \frac{1}{k} \cdot \text{sigmoid}(\hat{q}_c(x, A(k)) + \sum_{l \neq k} \frac{1}{|k - l|} \cdot \mathbb{W}_c(A(l), A(k)))$$

$$q_r(x, A(k)) = \hat{q}_r(x, A(k)) + \lambda \cdot \sum_{l \neq k} \frac{1}{|k - l|} \cdot \mathbb{W}_r(A(l), A(k)),$$

where $\mathbb{W}_c$ and $\mathbb{W}_r$ are sampled from a uniform distribution with range $[-3.0, 3.0]$ and $[-1.0, 1.0]$, respectively. Additionally, $\hat{q}_c(x, A(k))$ and $\hat{q}_r(x, A(k))$ are defined as follows.

$$\hat{q}_c(x, A(k)) = x^T M_{x, x_a}^c x_a + (\theta_x^c)^T \cdot x + (\theta_a^c)^T \cdot x_a,$$

$$\hat{q}_r(x, A(k)) = x^T M_{x, x_a}^r x_a + (\theta_x^r)^T \cdot x + (\theta_a^r)^T \cdot x_a,$$

where $x_a$ denotes action context for $A(k) = a$ represented by a one-hot vector. $M_{x, x_a}$, $\theta_x$, and $\theta_a$ are sampled from a uniform distribution with range $[-1, 1]$.

We then synthetically define a logging policy as in Eq. 12. We adopted a modified version of the Plackett–Luce based formulation in Eq. 12 because it allows a smooth transition between stochastic and deterministic regimes through the parameter. This design enables controlled experiments in which we can vary the determinism of the logging policy while holding other factors constant, making it ideal for studying bias behavior across regimes in our experiments. Similarly, the new policy in Eq. 13 follows a standard $\epsilon$-greedy form that allows adjustable stochasticity via $\epsilon$, a setup widely used in prior OPE studies Saito & Joachims (2022); Kiyohara et al. (2023).

**Additional Results.**  From Figure 7 to 9, we report additional results from the synthetic experiments to demonstrate that CIPS outperforms the baselines. We vary the new ranking policies, the definition of logging and new policies, and the noise on the click probability of CIPS. Note that the logged data size is set to 1000, the number of unique actions is 6, the ranking length is 6, $\alpha = \infty$, $\epsilon = 0.3$, and $\lambda = 0.5$.

**How does CIPS perform with varying new ranking policies?**  Figure 7 shows comparisons of estimators under varying new ranking policies ($\epsilon \in \{0.0, 0.2, 0.4, 0.6, 0.8, 1.0\}$ in Eq. 13). A larger value of $\epsilon$ makes the new ranking policy closer to a random uniform distribution. $\epsilon = 0.0$ means the policy is completely deterministic. We observe that CIPS outperforms the baselines across all values of $\epsilon$. As $\epsilon$ increases, the probability of the supported action increases. Consequently, the baselines reduce their bias. However, the baselines still exhibit severe bias, particularly when the new ranking policy is completely deterministic ($\epsilon = 0.0$).

**How does CIPS perform with different definition of logging and new policies?**  Figure 8 reports comparisons of estimators with a new different definition of logging and new policies from Eq. 12

and 13. For the logging policy, we define $f_0(x, a)$ in Eq. 12 as $f_0(x, a) = \hat{q}_c(x, A(k)) + \sigma$, where $\sigma$ is sampled from the standard normal distribution. We then construct the new policies by mixing the softmax and epsilon greedy policies as follows.

$$
\pi(A \mid x) = \prod_{k=1}^{K} 0.5 \times \left[ (1 - \epsilon) \cdot \mathbb{I}\left[ a(k) = \operatorname*{argmax}_{a' \in \mathcal{A} \setminus \{a_{1:k-1}\}} f_1(x, a') \right] + \frac{\epsilon}{|\mathcal{A} \setminus a_{1:k-1}|} \right]
$$
$$
+ 0.5 \times \frac{\exp(f_1(x, a(k))), \mathbb{I}[a(k) \notin a_{1:k-1}]}{\sum_{a' \in \mathcal{A} \setminus a_{1:k-1}} \exp(f_1(x, a'))}
$$

In Figure 8, we can see that CIPS outperforms the baselines under more complex definition of logging and new policies. This results suggest that the advantage of CIPS is not limited by the structures of logging and new polices.

**How does CIPS perform with varying noise on the click probability of CIPS?** In Figure 9, we compares CIPS against the baselines as we vary the accuracy of click probability used in CIPS. The x-axis represents levels of noise, which is sampled from a uniform distribution with range $[-\delta, \delta]$. Figure 9 shows that CIPS is superior to the baselines with noisy click probability, while its bias gradually increases. This increase of bias is consistent with Theorem 3.2. These results further demonstrate the robustness of CIPS even with larger noise on its click probability model and its consistent relative advantage over the baselines (note that the baselines are independent of the noise parameter).

From these results, we can see that it is important to use accurate click probabilities for low MSE estimations. Regarding model selection for estimating click probabilities, we can leverage data-driven estimator selection techniques, such as the ones proposed by Udagawa et al. (2023) and Felicioni et al. (2024). By treating CIPS with different click-probability estimators as separate estimators, these techniques allow us to automatically and data-drivenly identify the click-probability model that minimizes the resulting MSE.

### D.2 ABLATION STUDY ON SYNTHETIC DATA

In addition to the main performance comparisons, we conduct two ablation studies.

Figure 10 compares CDR against CIPS as we vary the accuracy of the regression model $\hat{q}_r$ used in CDR. Here, the x-axis represents the estimation error of the regression model; larger values correspond to less accurate estimates of $q_r$. The results show that, when $q_r$ is estimated with reasonable accuracy, CDR outperforms CIPS by leveraging its variance reduction effect, consistent with our theoretical analysis. We also observe that as the estimation noise in the regression model increases, the variance of CDR grows steadily, eventually eliminating its advantage over CIPS.

Next, Figure 11 evaluates the policy selection accuracy of the estimators under varying logged data sizes ($n$) and deterministic user thresholds ($\alpha$). We measure how accurately each estimator identifies the better policy between the new policy $\pi$ and the logging policy $\pi_0$ across 100 independent trials. Note that, in our setup, the new policy consistently outperforms the logging policy in expectation. The left plot demonstrates that CIPS achieves the highest selection accuracy across all data sizes, while the baselines fail entirely with zero accuracy. This failure is expected because, under severe common support violations, as shown in Theorem 2.1, baseline methods substantially underestimate the value of the new policy $\pi$, leading to systematic errors in policy selection. The right plot shows that CIPS maintains high selection accuracy even at larger $\alpha$ values, where the logging policy becomes more deterministic and the baselines deteriorate substantially. These findings highlight the superiority of CIPS over the baselines in both policy selection and policy evaluation.

### D.3 REAL-WORLD EXPERIMENTS

**Detailed Setup.** The detailed real-world experimental settings are described in the main text. We conduct OPE experiments on a real-world dataset called KuaiRec Gao et al. (2022), which consists of recommendation logs from the video-sharing app Kuaishou. KuaiRec contains fully observed user–item interactions with nearly 100% density for a subset of its users and items. By leveraging these user–item interactions, we can construct OPE experiments with minimal synthetic components Gao et al. (2022).

KuaiRec includes `user_feature.csv`, which we use as context vectors ($x$). We apply feature dimension reduction using PCA implemented in scikit-learn (Pedregosa et al., 2011). We then use the user–item interaction matrix recorded in the original data as the potential reward function $q_r(x, A(k))$. We construct $q_r(x, A(k))$ by randomly sampling rows and columns from the user–item matrix. We also define the expected click probability as

$$q_c(x, A(k)) = \begin{cases} 1-\eta_{x,A(k)} & \text{if } q_r(x, A(k)) > 2.0 \\ \eta_{x,A(k)} & \text{otherwise} \end{cases}, \tag{17}$$

where $\eta$ is a noise parameter sampled independently from the uniform distribution over $[0, 0.5]$. We set the threshold for binarization by following the example provided on the KuaiRec webpage. This definition reflects essential user behaviors, namely that more relevant items are more likely to be clicked, and that user responses remain probabilistic even for highly relevant items. Thus, this setting has some logic and is consistent with our formulations, and we have designed it with available modeling rationale in mind.

**Additional discussion about further bias mitigation.** In Figure 5, we observe that CIPS exhibits some non-negligible bias. This is because both the logging and new policies are deterministic, violating even the click-wise common support (Condition 3.1) of our methods. In this extreme case, no OPE method, including ours and existings, can remain strictly unbiased due to the absence of overlapping actions. Nevertheless, our results show that CIPS maintains significantly smaller bias and MSE than existing methods because it can still utilize some stochasticity embedded in user clicks. To further mitigate bias in the case with both deterministic logging and new policies, we can additionally leverage action embeddings when available as proposed in Saito & Joachims (2022), which allows us to use similarities across different actions in OPE.

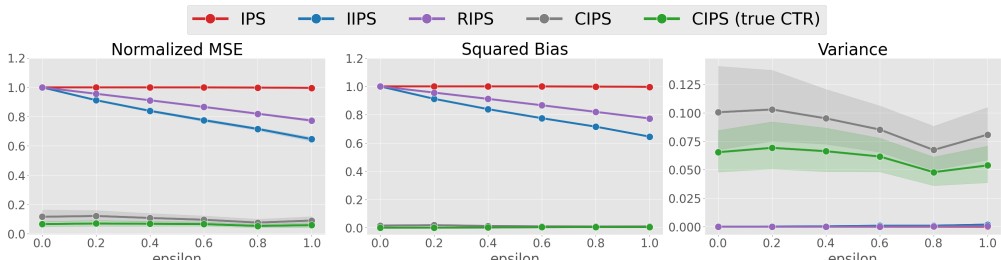

Figure 7: Comparison of MSE, Squared Bias, and Variance with varying new policies ($\epsilon$).

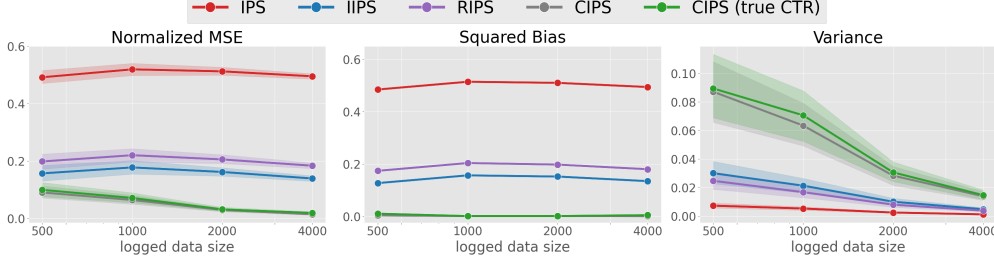

Figure 8: Comparison of MSE, Squared Bias, and Variance with different policy definition.

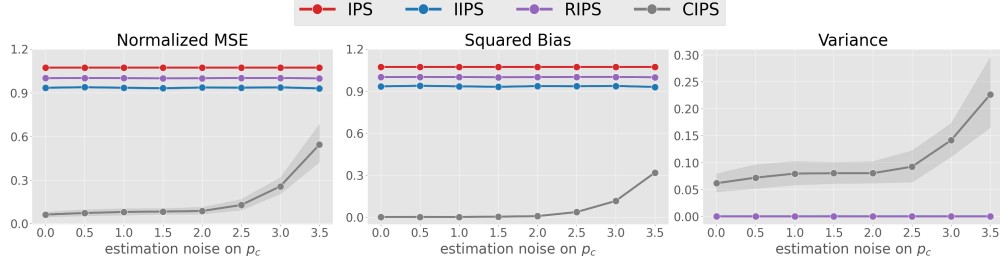

Figure 9: Comparison of MSE, Squared Bias, and Variance with varying estimation noise on $p_c$.

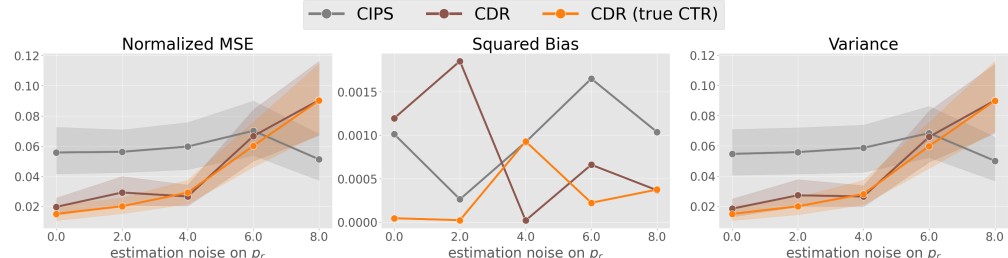

Figure 10: Comparison of CIPS and CDR' MSE, Squared Bias, and Variance with varying estimation noises on $q_r$.

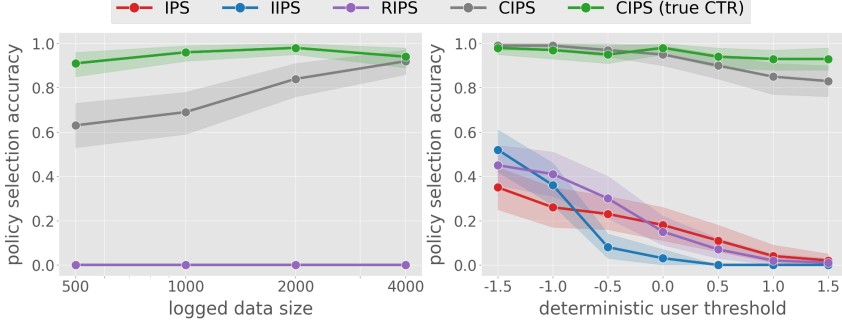

Figure 11: Comparison of the policy selection accuracy with varying logged data sizes ($n$) and deterministic user thresholds ($\alpha$).

