# OpenReview forum: "Off-Policy Evaluation for Ranking Policies under Deterministic Logging Policies"
_ICLR.cc/2026/Conference — ICLR 2026 Poster_

### Official Review · Reviewer_wdph · 2025-10-28

**Soundness:** 3
**Presentation:** 3
**Contribution:** 3
**Rating:** 6
**Confidence:** 4

**Summary:**

This paper studies off-policy evaluation (OPE) for ranking policies under deterministic logging, where existing estimators such as IPS, IIPS, and RIPS suffer from severe bias due to the lack of exploration. The authors propose a new estimator, Click-based Inverse Propensity Score (CIPS), which leverages the stochasticity of user click behavior instead of the logging policy. The paper provides theoretical analyses proving that CIPS is unbiased under mild assumptions and demonstrates through synthetic and real-world experiments that CIPS substantially reduces bias compared to existing methods.

**Strengths:**

1. The paper addresses an important and realistic problem in OPE that has been largely overlooked — deterministic logging policies.
2. The idea of exploiting user click stochasticity as a new source of importance weighting is conceptually novel.
3. Theoretical results are clearly stated and well justified.
4. The paper is clearly written and easy to follow.

**Weaknesses:**

1. The effectiveness of CIPS depends on the accuracy of estimated click probabilities, which may be difficult to obtain in real systems.
2. The independence of potential rewards assumption may not hold in many practical ranking scenarios.
3. Real-world validation relies on simulated click probabilities (Eq. 13), limiting the strength of empirical evidence.
4. No statistical significance or confidence intervals are reported for the experimental results.

**Questions:**

1. Could the authors clarify the rationale behind the reward design, particularly the two-stage structure of “clicks” and “potential rewards”? How is this modeling choice connected to real-world ranking scenarios, and to what extent does it affect the theoretical guarantees or the empirical behavior of CIPS? For example, would CIPS still perform similarly if the downstream reward R were not conditioned on the click C?
2. In the real-world experiments (KuaiRec), the click probabilities were generated using a simulated noise model (Eq. 13).
Could the authors provide more discussion or evidence about how well this setting reflects real deterministic logging systems?
3. While CIPS focuses on bias reduction, could the authors elaborate on its variance behavior in practical settings where click probabilities are estimated with noise? How sensitive is CIPS to misspecification of the click model?

---

> ### Author Response · Authors · 2025-11-18
>
> We appreciate the thoughtful and valuable feedback from the reviewer. We will respond to the specific comments and questions below.
>
> > Could the authors clarify the rationale behind the reward design, particularly the two-stage structure of “clicks” and “potential rewards”? How is this modeling choice connected to real-world ranking scenarios, and to what extent does it affect the theoretical guarantees or the empirical behavior of CIPS? For example, would CIPS still perform similarly if the downstream reward R were not conditioned on the click C?
>
> We appreciate the reviewer’s question about our two-stage formulation of clicks $C$ and potential rewards $R$. This structure is based on real-world ranking systems, such as Amazon product search, YouTube video recommendation, and many other large-scale e-commerce or content platforms, where user interactions naturally follow two or more sequential phases, i.e., exposure and click, followed by downstream engagement (e.g., purchase, watch time, or dwell time). We believe it is actually quite difficult to find concrete, practical ranking applications that do not involve two or more stages in their reward design.
>
> It is admittedly hard to imagine such a practical ranking scenario, but if there were no click behavior before the reward is observed, then it is true that we could no longer leverage the additional stochasticity provided by clicks. In this case, both CIPS and existing methods would be expected to suffer from bias due to deterministic logging. For such settings, additional modeling mechanisms, such as action embeddings or side-information generalization, as explored by Saito and Joachims (2022) and Felicioni et al. (2022), would be effective approaches for mitigating the issues caused by deterministic logging without click signals. **While this is beyond the scope of our current work, we agree that extending CIPS by integrating action embeddings to handle even more challenging deterministic-logging scenarios without click logs would be an interesting future direction.**
>
> ---
>
> (Saito and Joachims (2022))   Yuta Saito, Thorsten Joachims. Off-Policy Evaluation for Large Action Spaces via Embeddings. https://arxiv.org/abs/2202.06317
>
> (Felicioni et al. (2022))Nicolò Felicioni, Maurizio Ferrari Dacrema, Marcello Restelli, Paolo Cremonesi. Off-Policy Evaluation with Deficient Support Using Side Information. https://proceedings.neurips.cc/paper_files/paper/2022/hash/c32be49c09eec3aad1f2bb587543e7f6-Abstract-Conference.html
>
> > In the real-world experiments (KuaiRec), the click probabilities were generated using a simulated noise model (Eq. 13). Could the authors provide more discussion or evidence about how well this setting reflects real deterministic logging systems?
>
> Thank you for the comment. Equation (13) is a simplified model to simulate realistic user click stochasticity. Its form, $p_c(x, A(k)) = \text{sigmoid}(q_r(x, A(k)) + \eta)$ with $\eta \sim U[0,0.5] $, reflects two essential user behaviors.
>
> - more relevant items are more likely to be clicked, and
> - user responses remain probabilistic even for highly relevant items.
>
> As explained above, this setting has some logic and is consistent with our formulations, and we have designed it with available modeling rationale in mind. However, it is also true that the true form of real-world reward functions is fundamentally unknown, and thus it is difficult to provide stronger evidence beyond this level of abstraction. Strictly speaking, the ideal way to validate an offline evaluation method would be to conduct production A/B testing after the offline evaluation and verify its predictive accuracy against online results. Yet, neither our work nor most existing OPE studies (including those published at ICLR and related venues) have access to such controlled online validation. **We acknowledge that this is a shared limitation of the entire OPE community, and we believe that accumulating empirical knowledge through collaborative industrial studies is an important next step for the field.** We will clarify this as a future direction of our work.

---

> > ### Author Response · Authors · 2025-11-18
> >
> > > While CIPS focuses on bias reduction, could the authors elaborate on its variance behavior in practical settings where click probabilities are estimated with noise? How sensitive is CIPS to misspecification of the click model?
> >
> > Thank you for the questions, we are happy to clarify them. While CIPS primarily targets bias reduction, Theorem 3.3 analyzes its variance. We demonstrate that variance scales with the magnitude of click-based weights $w(x,a,\pi,\pi_0)$, which remain stable in deterministic logging (unlike IPS, where weights explode). Empirically, CIPS exhibits moderate variance, as seen in Figs. 1 - 4. If needed, it is straightforward to extend our estimators by including weight clipping, self-normalization, or other recent variance reduction techniques (such as the use of action embeddings) to further improve, which could be an interesting future direction.
> >
> > Regarding robustness against the misspecification of the click model. **Both our synthetic and real-world experiments demonstrate that, although our estimators exhibit a small additional bias when using estimated click probabilities, CIPS consistently outperforms all existing methods, even under estimated click models.** This is because the bias caused by deterministic logging in existing OPE estimators is far more severe than the minor estimation bias introduced in CIPS.
> >
> > **To further address the reviewer’s concern, we additionally conducted an experiment to test the robustness of CIPS against estimation errors in the click-probability estimator.** Specifically, we gradually add synthetic uniform noise to the click-probability estimator used by CIPS and compare its resulting MSE with the baselines. The results are shown in the table below.
> >
> > | uniform noise parameter |   0 | 1 | 2 | 3 |
> > |:--------------------------|--------:|--------:|--------:|--------:|
> > | IPS  | 1.0722 | 1.0722 | 1.0722 | 1.0722 |
> > | IIPS  | 0.9344 | 0.9344  | 0.9344  | 0.9344 |
> > | RIPS | 1.0002 | 1.0002 | 1.0002 | 1.0002 |
> > | CIPS | **0.0617** | **0.0796**  | **0.0863** | **0.2559** |
> >
> > **These results further demonstrate the robustness of CIPS even with larger noise on its click probability model and its consistent relative advantage over the baselines** (note that the baselines are independent of the noise parameter). We will add more details about this additional experiment in the revision.
> >
> > Moreover, in practice, we can leverage data-driven estimator selection techniques, such as the ones proposed by Udagawa et al. (2021) and Felicioni et al. (2024) to perform model selection of the click model. By treating CIPS with different click-probability estimators as separate estimators, these techniques allow us to automatically identify the click-probability model that minimizes the MSE in the downstream OPE task. We will include this important discussion in the revised version.
> >
> > ---
> >
> > (Udagawa et al. (2021)): Takuma Udagawa, Haruka Kiyohara, Yusuke Narita, Yuta Saito, Kei Tateno. Policy-Adaptive Estimator Selection for Off-Policy Evaluation. https://arxiv.org/abs/2211.13904
> >
> > (Felicioni et al. (2024)): Nicolò Felicioni, Michael Benigni, Maurizio Ferrari Dacrema. Automated Off-Policy Estimator Selection via Supervised Learning. https://arxiv.org/abs/2406.18022
> >
> > > No statistical significance or confidence intervals are reported for the experimental results.
> >
> >
> > **We would like to clarify that all reported empirical results already include 95% bootstrap confidence intervals in the figures.** The intervals were estimated over 100 independent random seeds using resampling on evaluation metrics. Moreover, our released code enables full reproducibility of these intervals and all experimental statistics. We will clarify this explicitly in the revised text and figure captions.

---

### Official Review · Reviewer_RmhY · 2025-10-31

**Soundness:** 2
**Presentation:** 2
**Contribution:** 3
**Rating:** 4
**Confidence:** 3

**Summary:**

This paper investigates the severe bias introduced by existing Off-Policy Evaluation (OPE) methods when applied under deterministic logging, and illustrates this problem through concrete examples and theoretical analysis. To address this issue, the authors propose a Click-based Inverse Propensity Score (CIPS) estimator, which exploits the intrinsic stochasticity of user click behavior through click importance weights, substantially relaxing the support condition and enabling unbiased or low-bias estimation under deterministic logging. The effectiveness of the proposed method is thoroughly validated through rigorous theoretical analysis that establishes its unbiasedness and consistency properties, and extensive experimental results that empirically confirm its superior performance over existing OPE approaches under deterministic logging.

**Strengths:**

1. The paper is motivated by a well-founded research question — existing OPE methods are primarily developed for stochastic logging policies, which inherently limit their reliability under deterministic settings. This motivation is both sound and clearly articulated.
2. To alleviate the bias commonly induced by the OPE methods under deterministic logging, the paper relaxes the stochasticity assumption of existing OPE methods and takes a novel perspective by leveraging the inherent randomness in user click behaviors. By leveraging click probabilities for reweighting, the proposed CIPS method represents a clever and conceptually elegant reformulation of the problem.
3. The paper clearly illustrates the bias issue of IPS-based estimators under deterministic logging through intuitive examples, making the problem easy to understand. Moreover, the theoretical analysis is thorough and well-justified, and the experimental results further validate that CIPS effectively mitigates bias across varying degrees of determinism in logging policies.

**Weaknesses:**

1. The CIPS method implicitly assumes consistency in the action space between the old and new policies. It remains unclear whether CIPS can maintain its robustness when new actions appear under the new policy.
2. In the synthetic data section, the exact forms of functions appearing in Equations (9) and (10) are not clearly specified.
3. The paper lacks a dedicated related work section. The relation between this work and related work should be discussed.
4. Baselines are limited. More recent baselines should be used. The strongest baseline, RIPS, was proposed in 2020; it is unclear whether more recent OPE methods should be considered for comparison.

**Questions:**

1. In real-world scenarios, although the IPS exhibits relatively large bias compared to other approaches, why does this bias remain unchanged when the logging policy shifts from stochastic to more deterministic?
2. In real-world scenarios, why does the CIPS method appear less robust to bias across varying degrees of deterministic policies compared to synthetic data? What might be the underlying causes of this discrepancy?

---

> ### Author Response · Authors · 2025-11-18
>
> We appreciate the thoughtful and valuable feedback from the reviewer. We will respond to the specific comments and questions below.
>
> > The CIPS method implicitly assumes consistency in the action space between the old and new policies. It remains unclear whether CIPS can maintain its robustness when new actions appear under the new policy.
>
> **We would like to point out the potential misunderstanding made by the reviewer. Handling new actions (i.e., items completely unseen under the logging policy) represents a different research problem and motivation, which lies outside the scope of our work as well as all existing ranking OPE studies (Li et al., 2018; McInerney et al., 2020).** Our motivation is NOT to deal with the existence of new actions, but rather to deal with deterministic logging, and thus we do not care about the robustness to new actions in our research.
>
> **Nonetheless, we know how to potentially deal with new actions in practical scenarios.** If one aims to evaluate new actions beyond logged support, additional modeling mechanisms such as action embeddings or side-information generalization, as explored by Saito and Joachims (2022) and Felicioni et al. (2022), would be required. It is straightforward to extend our method to incorporate such additional structure to potentially deal with new actions when needed in practice. **While we agree that new actions are practically relevant, this issue is orthogonal to our contribution and does not challenge the validity or robustness of CIPS within its intended problem scope.**
>
> ---
> (Saito and Joachims (2022))   Yuta Saito, Thorsten Joachims. Off-Policy Evaluation for Large Action Spaces via Embeddings. https://arxiv.org/abs/2202.06317
>
> (Felicioni et al. (2022))Nicolò Felicioni, Maurizio Ferrari Dacrema, Marcello Restelli, Paolo Cremonesi. Off-Policy Evaluation with Deficient Support Using Side Information. https://proceedings.neurips.cc/paper_files/paper/2022/hash/c32be49c09eec3aad1f2bb587543e7f6-Abstract-Conference.html
>
>
>
> > In the synthetic data section, the exact forms of functions appearing in Equations (9) and (10) are not clearly specified.
>
>
> We appreciate the reviewer’s request for more clarity. It is true that we describe only the high-level formulations of the synthetic reward functions in Eq.(9) and (10), **their exact formulations and parameters are, in fact, already clearly described in Appendix E.1 and included in the released code, which ensures full reproducibility of all synthetic experiments.**
>
>
> > The paper lacks a dedicated related work section. The relation between this work and related work should be discussed.
>
>
> **We respectfully note that the claim about a missing related work section is incorrect.
> A detailed discussion of related work is already provided in Appendix A,** covering classical OPE methods (Dudík et al., 2014), ranking-specific estimators (Li et al., 2018; McInerney et al., 2020), and recent relevant extensions, such as the use of action embeddings in general OPE (Saito and Joachims., 2022). To avoid confusion, we will explicitly reference Appendix A in the main text so that readers can easily locate this section.
>
>
> > Baselines are limited. More recent baselines should be used. The strongest baseline, RIPS, was proposed in 2020; it is unclear whether more recent OPE methods should be considered for comparison.
>
> **We appreciate the reviewer’s concern, but we respectfully disagree that the age of a baseline (e.g., RIPS, 2020) implies insufficiency. The relevant baselines for this research direction are determined by methodological relevance, not publication year.** Our study exhaustively includes IPS methods directly applicable to ranking OPE under stochastic logging, i.e., IPS, IIPS, and RIPS, which remain the state-of-the-art for this problem.
>
> **Furthermore, none of the other reviewers raised concerns regarding our baseline selection. If the reviewer believes that additional, more recent methods should be included, we kindly ask them to specify which ones and clarify their methodological relevance to deterministic ranking OPE, as well as the concrete reasons they should be compared beyond the current baselines.** We would greatly appreciate learning the reviewer’s thoughtful perspective on baseline selection, as it could help us further enhance the value of our experimental evaluation. However, if there are no specific suggestions or rationales, we believe that the baselines we have selected are appropriate and sufficient for assessing the proposed method.

---

> > ### Author Response · Authors · 2025-11-26
> > **We would appreciate knowing whether our responses have sufficiently addressed the reviewer’s initial concerns.**
> >
> > Thank you again for taking the time to carefully review and discuss our paper. **In our responses, we believe we have addressed all concerns raised in the initial review. At this point, we are not aware of any remaining issues that would prevent an updated assessment. If there are still concerns that motivate the current score of 4, we would greatly appreciate the opportunity to understand them more fully, which we believe would be highly valuable for further improving the paper**. We would be grateful if the reviewer could clarify any remaining issues soon, especially as the discussion deadline is approaching.

---

### Official Review · Reviewer_m2rz · 2025-11-01

**Soundness:** 3
**Presentation:** 3
**Contribution:** 3
**Rating:** 6
**Confidence:** 3

**Summary:**

This paper addresses the challenge of Off-Policy Evaluation (OPE) for ranking policies, which aims to estimate a new policy's performance using data collected by a different, existing policy. The authors highlight that standard estimators, like Inverse Propensity Score (IPS), fail and suffer from severe bias when the logging policy is deterministic.
The paper's contribution is an estimator called the Click-based Inverse Propensity Score (CIPS). The key finding is that reliable OPE is possible even with deterministic logging by exploiting a different source of randomness: the intrinsic stochasticity of user click behavior. Instead of weighting by the policy's action probabilities, CIPS uses the ratio of marginalized click probabilities as a new form of importance weighting. Theoretical analysis shows CIPS is unbiased under specific conditions. Experiments on synthetic and real-world data demonstrate that CIPS achieves lower bias and more accurate estimations than existing methods in deterministic settings.

**Strengths:**

* The paper's primary strength is proposing an approach to address the Off-Policy Evaluation (OPE) issue when the data-logging policy is fully deterministic, a setting where existing methods fail.

* The CIPS estimator method interestingly shifts the source of randomness away from the deterministic policy and instead exploits the intrinsic stochasticity of user click behavior for importance weighting. This approach is shown to "significantly" reduce the severe bias of standard estimators in this challenging but practical setting.

**Weaknesses:**

The following are some of the items that it make the paper better to see them developed/explained/addressed further:

* The theoretical unbiasedness of CIPS depends on Condition 3.2, which assumes that the expected potential reward for an item (e.g., a purchase after a click) depends only on that item, not on the other items shown in the ranking. The paper's own experiments show that as this condition is more strongly violated, the Mean Squared Error (MSE) of CIPS, while still the best, does increase. The experiment section didn’t develop much on this aspect and how to potentially address it.

* The CIPS estimator relies on knowing the true click probabilities for both the logging and new policies. In practice, these are unknown and must be estimated from data. As Theorem 3.2 shows, any inaccuracy in the ratio of these estimated click probabilities will introduce bias into the final policy evaluation.

* While CIPS is designed for deterministic logging policies, it can still suffer from "non-negligible bias" in the most challenging scenarios. Specifically, the real-world experiment (which used a deterministic new policy in addition to a deterministic logging policy) showed this bias. The authors state this is due to violations of the click-wise common support (Condition 3.1), which can happen when both policies are deterministic. It would be great to develop further how to address this.

**Questions:**

1- Your theoretical analysis for CIPS's unbiasedness relies on the "Independence of Potential Rewards" (Condition 3.2). How robust is the CIPS estimator in practical scenarios where this assumption is strongly violated, such as systems with significant item complementarity or competition, and what is the expected impact on its bias?

2- Theorem 3.2 indicates that the bias of CIPS depends on the accuracy of the ratio of estimated click probabilities. How sensitive is the estimator's performance to errors in this ratio, particularly in low-data regimes, and what are the practical implications for model selection when estimating these click probabilities from logged data?

3- Your real-world experiment noted non negligible bias when both the logging and new policies were deterministic. This suggests violations of the clickwise common support (Condition 3.1). Could you elaborate on the performance degradation in this specific scenario and discuss potential extensions or clipping methods to mitigate this bias when even clickbased support is deficient?

---

> ### Author Response · Authors · 2025-11-18
>
> We appreciate the thoughtful and valuable feedback from the reviewer. We will respond to the specific comments and questions below.
>
> > Your theoretical analysis for CIPS's unbiasedness relies on the "Independence of Potential Rewards" (Condition 3.2). How robust is the CIPS estimator in practical scenarios where this assumption is strongly violated, such as systems with significant item complementarity or competition, and what is the expected impact on its bias?
>
> Thank you for raising the important point to discuss. We agree with the reviewer that Condition 3.2 may not strictly hold in systems with strong item complementarity or competition. **However, we would like to emphasize that Condition 3.2 is considerably milder than the independence assumptions required by existing estimators (e.g., Li et al., 2018). Thus, in scenarios where this assumption is strongly violated, CIPS and existing methods would both be affected, and CIPS would at least perform better than the existing method.**
>
> Empirically, we directly tested this by increasing the level of inter-ranking interactions in our synthetic experiments (Figure 3). **While the bias of CIPS increased slightly with stronger violations of the condition, the bias induced by deterministic logging in existing IPS-based estimators was an order of magnitude larger. Across all tested conditions, CIPS consistently outperformed every baseline, never yielding inferior results even when independence was severely violated.** Overall, we agree with the reviewer that it is difficult to predict the absolute effect of the condition violation on CIPS, as this is highly case-dependent. However, both empirically and theoretically, the relative advantage of CIPS over existing baselines remains unchanged, and this is what we can and did demonstrate in our research. We will make this point explicit in the revised version.
>
>
> > Theorem 3.2 indicates that the bias of CIPS depends on the accuracy of the ratio of estimated click probabilities. How sensitive is the estimator's performance to errors in this ratio, particularly in low-data regimes, and what are the practical implications for model selection when estimating these click probabilities from logged data?
>
> This is an important point to discuss. We would like to clarify that both the synthetic and real-world experiments already demonstrate that CIPS remains effective even when click probabilities are estimated using standard prediction models. **Particularly, in synthetic experiments, CIPS computed with a learned click probability model (based on neural networks) exhibited only marginal bias relative to the oracle CIPS using true click probabilities for a range of logged data sizes including low-data ones (Figure 1).**
>
> To further address the reviewer’s concern, we additionally conducted an experiment to test the robustness of CIPS against estimation errors in the click-probability estimator. Specifically, we gradually add synthetic uniform noise to the click-probability estimator used by CIPS and compare its resulting MSE with the baselines. The results are shown in the table below.
>
> | uniform noise parameter |   0 | 1 | 2 | 3 |
> |:--------------------------|--------:|--------:|--------:|--------:|
> | IPS  | 1.0722 | 1.0722 | 1.0722 | 1.0722 |
> | IIPS  | 0.9344 | 0.9344  | 0.9344  | 0.9344 |
> | RIPS | 1.0002 | 1.0002 | 1.0002 | 1.0002 |
> | CIPS | **0.0617** | **0.0796**  | **0.0863** | **0.2559** |
>
> These results further demonstrate the robustness of CIPS even with larger noise on its click probability model and its consistent relative advantage over the baselines (note that the baselines are independent of the noise parameter). We will add more details about this additional experiment in the revision.
>
> Regarding model selection for estimating click probabilities, we can leverage data-driven estimator selection techniques, such as the ones proposed by Udagawa et al. (2021) and Felicioni et al. (2024). By treating CIPS with different click-probability estimators as separate estimators, these techniques allow us to automatically and data-drivenly identify the click-probability model that minimizes the resulting MSE. We will include this important discussion in the revised version.
>
> ---
>
> (Udagawa et al. (2021)): Takuma Udagawa, Haruka Kiyohara, Yusuke Narita, Yuta Saito, Kei Tateno. Policy-Adaptive Estimator Selection for Off-Policy Evaluation. https://arxiv.org/abs/2211.13904
>
> (Felicioni et al. (2024)): Nicolò Felicioni, Michael Benigni, Maurizio Ferrari Dacrema. Automated Off-Policy Estimator Selection via Supervised Learning. https://arxiv.org/abs/2406.18022

---

> > ### Author Response · Authors · 2025-11-18
> >
> > > Your real-world experiment noted non negligible bias when both the logging and new policies were deterministic. This suggests violations of the clickwise common support (Condition 3.1). Could you elaborate on the performance degradation in this specific scenario and discuss potential extensions or clipping methods to mitigate this bias when even clickbased support is deficient?
> >
> > Thank you so much for the valuable comment. As the reviewer correctly points out, residual bias may arise when both the logging and new policies are deterministic, violating even the click-wise common support (Condition 3.1) of our methods. In this extreme case, no OPE method, including ours and existings, can remain strictly unbiased due to the absence of overlapping actions. **Nevertheless, our results show that CIPS maintains significantly smaller bias and MSE than existing methods because it can still utilize some stochasticity embedded in user clicks.**
> >
> >
> > To further mitigate bias in the case with both deterministic logging and new policies, we can additionally leverage action embeddings when available as proposed in Saito and Joachims (2022), which allows us to use similarities across different actions in OPE. While the integration of action embeddings are out of scope of our paper, it would be worth mentioning as an interesting future direction, and we will in our revision.
> >
> > ---
> >
> > (Saito and Joachims (2022))   Yuta Saito, Thorsten Joachims. Off-Policy Evaluation for Large Action Spaces via Embeddings. https://arxiv.org/abs/2202.06317

---

> > ### Comment · Reviewer_m2rz · 2025-11-26
> >
> > Thank you for the clarification and addressing my concerns. I'll keep my rating as I am welcoming the acceptance of this paper.

---

> > > ### Author Response · Authors · 2025-11-26
> > >
> > > We would like to thank the reviewer once again for their thoughtful review and for confirming that our responses have addressed all of their concerns.

---

### Official Review · Reviewer_WNN2 · 2025-11-03

**Soundness:** 3
**Presentation:** 3
**Contribution:** 3
**Rating:** 6
**Confidence:** 4

**Summary:**

This paper studies the off-policy evaluation for ranking policies. Existing approaches require the data collection (logging) policy to be sufficiently stochastic and do not perform well when it is deterministic. This paper has overcome this challenge by proposing novel estimators, referred to as Click-based Inverse Propensity Score (CIPS), based on the intrinsic stochasticity of user click behavior. This paper proceeds as follows: the considered off-policy evaluation problem is formulated in Section 2, and the CIPS estimator is proposed and analyzed in Section 3. Preliminary experimental results are demonstrated in Section 4.

**Strengths:**

- This paper has considered a very practical problem: how to do off-policy evaluation for ranking policies when the data collection (logging) policy is deterministic (or not sufficiently stochastic). This is a practical and important problem in algorithmic ranking systems, such as recommendation systems.

- The proposed algorithm, CIPS, is natural and simple.

Overall, I think this is an interesting paper, and recommend accepting it.

**Weaknesses:**

- I have concerns about some key math notations in this paper. In particular, notations like $C(a)$, $R(a)$, $C(k)$, and $R(k)$ can be misleading. Specifically, $C(k)$ hints that the click event **only** depends on the position $k$, while $C(a)$ hints that the click event **only** depends on the action $a$. I do not think this is what this paper has assumed. I recommend that the authors clean up such notations.

- The idea of this paper is very interesting. However, given the idea and the CIPS algorithm, the analyses for the CIPS algorithm (Theorems 3.1-3.3) are relatively straightforward based on existing literature and techniques. If the authors disagree, please explain the key challenges and novelties in the proofs for Theorems 3.1-3.3, compared to existing literature.

- In the experiments, this paper has only considered a fixed logging policy defined in equation 11 and a fixed "new" policy defined in equation 12. First, please better motivate and explain why choosing such a logging policy and such a new policy. Second, I strongly recommend that the authors also include experimental results with a different logging policy and a different new policy to justify that the advantages of CIPS are not limited to such policies.

- In Section 4.2, the click probability model for real-world data defined in equation 13 seems pretty arbitrary. Please better motivate and justify it.

**Questions:**

- Please try to address the weaknesses listed above. I will consider increasing my score if some of them are addressed.

- If possible, I recommend moving the Click-based Doubly Robust estimator to the main body of this paper. I think it is important.

---

> ### Author Response · Authors · 2025-11-18
>
> We appreciate the thoughtful and valuable feedback from the reviewer. We will respond to the specific comments and questions below.
>
> > I have concerns about some key math notations in this paper. In particular, notations like $C(a)$, $R(a)$, $C(k)$, and $R(k)$ can be misleading.
>
> Thank you very much for the valuable feedback. **In fact, our notation does not impose any independence assumption, i.e., it can generally depend on the actions presented at different positions.** For example, the independence assumption used in the IIPS estimator can be written as $ E[C(a)R(a) | A] = E[C(a)R(a) | a]  $, which shows that our notation itself does not imply independence. Nevertheless, we appreciate this insightful comment and will clarify the intended meaning of our notation in the revised version.
>
> > The idea of this paper is very interesting. However, given the idea and the CIPS algorithm, the analyses for the CIPS algorithm (Theorems 3.1-3.3) are relatively straightforward based on existing literature and techniques.
>
> Thank you for this insightful comment. We essentially agree with the reviewer that our theoretical analyses do not introduce particularly complex mathematical innovations. **However, we would like to emphasize that the main goal of our paper is not to pursue mathematical difficulty for its own sake. Rather, our objective is to develop practical and theoretically grounded methods that can effectively handle deterministic logging policies, an area that has remained unsolved in ranking OPE. In this sense, we do not view the simplicity of our derivations as a weakness; instead, we consider it a strength that our proposed methods address a previously intractable problem without requiring unnecessary or complicated additional mechanisms.**
>
> > In the experiments, this paper has only considered a fixed logging policy defined in equation 11 and a fixed "new" policy defined in equation 12. First, please better motivate and explain why choosing such a logging policy and such a new policy. Second, I strongly recommend that the authors also include experimental results with a different logging policy and a different new policy to justify that the advantages of CIPS are not limited to such policies.
>
> We appreciate the reviewer’s suggestion to clarify our policy design choices. Our primary experimental goal was to systematically control the degree of stochasticity versus determinism in the logging policy, since the central contribution of our paper is to improve OPE methods under deterministic logging.
>
> To this end, we adopted a modified version of the Plackett–Luce based formulation in Eq. (11) because it allows a smooth transition between stochastic and deterministic regimes through the parameter $\alpha$. This design enables controlled experiments in which we can vary the determinism of the logging policy while holding other factors constant, making it ideal for studying bias behavior across regimes in our experiments. Similarly, the new policy in Eq. (12) follows a standard ε-greedy form that allows adjustable stochasticity via $\epsilon $, a setup widely used in prior OPE studies, e.g. Kiyohara et al. (2023) and Saito and Joachims (2022).
>
> **Nonetheless, we agree that testing with alternative policies would further validate the generality of our method. We have already conducted additional experiments using different logging and new policies. We share the results of the additional experiment in the following table, showing consistent and substantial advantages of CIPS in terms of MSE.**
>
> | logged data sizes |    500 | 1000 | 2000 | 4000 |
> |:--------------------------|--------:|--------:|--------:|--------:|
> | IPS  | 0.4915 | 0.5191 | 0.5122 | 0.4950 |
> | IIPS  | 0.1566 | 0.1775  | 0.1617  | 0.1389 |
> | RIPS | 0.1982 | 0.2201 | 0.2053 | 0.1834 |
> | CIPS | **0.0903** | **0.0641**  | **0.0293** | **0.0141** |
>
> where we use completely different logging and new policies compared to the original experimental design we employed in the draft, showing robustness of our method. We will add more details about this additional experiment, including the definitions of the policies we used, in the revision.

---

> > ### Author Response · Authors · 2025-11-18
> >
> > > In Section 4.2, the click probability model for real-world data defined in equation 13 seems pretty arbitrary. Please better motivate and justify it.
> >
> > Thank you for the comment. Equation (13) is a simplified model to simulate realistic user click stochasticity. Its form, $p_c(x, A(k)) = \text{sigmoid}(q_r(x, A(k)) + \eta)$ with $\eta \sim U[0,0.5] $, reflects two essential user behaviors.
> >
> > - more relevant items are more likely to be clicked, and
> > - user responses remain probabilistic even for highly relevant items.
> >
> > As explained above, this setting has some logic and is consistent with our formulations, and we have designed it with available modeling rationale in mind. However, it is also true that the true form of real-world reward functions is fundamentally unknown, and thus it is difficult to provide stronger evidence beyond this level of abstraction. Strictly speaking, the ideal way to validate an offline evaluation method would be to conduct production A/B testing after the offline evaluation and verify its predictive accuracy against online results. Yet, neither our work nor most existing OPE studies (including those published at ICLR and related venues) have access to such controlled online validation. **We acknowledge that this is a shared limitation of the entire OPE community, and we believe that accumulating empirical knowledge through collaborative industrial studies is an important next step for the field.** We will clarify this as a future direction of our work.
> >
> >
> > >  If possible, I recommend moving the Click-based Doubly Robust estimator to the main body of this paper.
> >
> > This is a very useful comment, and we will try to put the descriptions of the CDR estimator in the revised version, leveraging an additional page allowed upon publication.

---

### Author Response · Authors · 2025-12-02
**Paper Revision**

Dear Reviewers,

Thank you again for the constructive and insightful feedback. We have revised the paper and all main points raised by the reviewers have now been addressed. Below we summarize the key updates, which are marked in red in the revised manuscript.

### 1. Clarifications of notation, assumptions, and theory
- Added an explicit explanation that notation such as $C(a), R(a), C(k), R(k)$ does not imply independence and clarified the intended meaning.
- Clarified the mildness of the “Independence of Potential Rewards” condition and added discussion of how CIPS behaves when it is violated.
- Added a clearer explanation of CIPS variance (Theorem 3.3) and noted standard extensions (weight clipping, self-normalization).
### 2. Additional experiments
- We included two new experiments directly addressing reviewer concerns in the appendix
    - Different logging / new policies: Added a new experimental setting with entirely different policy families, showing CIPS consistently outperforms IPS/IIPS/RIPS.
    - Noise robustness: Added results where synthetic noise is added to the click-probability model. CIPS remains stable and maintains a large advantage over baselines.
- Clarified that all figures already report 95% bootstrap confidence intervals.
### 3. Improved explanations of experimental design
- Added motivation for the logging policy (Eq. 11) and new policy (Eq. 12), explaining how they allow controlled transitions between stochastic and deterministic settings in the appendix.
- Added justification of the click probability model in Eq. 13 and clarified its role as a realistic but simplified model corresponding to our formulation in the appendix.
- Clearly pointed to Appendix E.1 for exact definitions of the synthetic reward functions.
### 4. Structural and presentation improvements
- Moved the CDR estimator description from the appendix to the main text.
- Clarified the location of the related work section (Appendix A) and added main-text references to it.
- Strengthened the explanation of the two-stage reward structure (click + downstream reward), connecting it to real-world ranking systems.
### 5. Discussion of deterministic logging, deficient support, and new actions
- Added explanation of CIPS’s behavior when both logging and new policies are deterministic and how click-level stochasticity still mitigates bias.
- Added future-direction notes on integrating action embeddings or side information to handle new or unseen actions (Saito & Joachims 2022).

**We believe these updates fully resolve the reviewers’ main concerns and significantly strengthen the manuscript. Thank you again for your thoughtful and valuable feedback.**

---

### Meta-Review · Area_Chair_R1K7 · 2026-01-07

**Summary:**

This paper studies off-policy evaluation (OPE) for ranking policies. The authors address an important and highly practical gap in OPE for ranking: existing approaches assume the data logging policy to be sufficiently stochastic, and suffer from bias during computing importance weights when the logging policy is deterministic. The authors propose Click-based Inverse Propensity Score (CIPS) and Click-based Doubly Robust (CDR) estimators to bridge the gap. Instead of weighting by the policy's action probabilities, the proposed estimators uses the ratio of marginalized click probabilities as a new form of importance weighting. Theoretical analysis shows CIPS and CDR are unbiased under certain assumptions. Experiments on synthetic and real-world data demonstrate the effectiveness of the estimators.

Three reviewers were initially positive about the paper, recognizing the problem as practical and the solution as natural and intuitive. The reviewers raised the following concerns: 1) independence of potential rewards assumption may not hold in practice, which is clarified in the revision;  2) robustness against noise data, which is addressed by new experiments on noisy clicks and the estimator still performed well; 3) limited to fixed logging and new policy, which is addressed by new experiments on different policies. The authors also made efforts in clarifying settings, notations, and experimental designs. Overall, most of the concerns are resolved in rebuttal and AC recommends acceptance of the paper.

**Reviewer Concerns:**

Noise Robustness: Reviewers wdph and m2rz were concerned about the sensitivity click noise.  The authors reported new experiments in the rebuttal showing that with noise added to the click-model, CIPS still outperforms baselines.

Fixed policies (WNN2): the authors provided additional results using different logging and new policies.

Reviewer RmhY's misunderstanding about missing related work section and missing modern baselines are clarified by author responses.

Independence of potential rewards assumption may not hold in practice: Reviewer m2rz  and wdph shared this question which is clarified from empirical perspective; this is the only potential "outstanding" concern that may benefit from more investigations.

**Reviewer Scores:**

Reviewer WNN2 (6): Reviewer mention willing to increase the score if the concerns were addressed, which AC believes is the case.

Reviewer m2rz (6): Confirmed keeping the positive score.

Reviewer wdph (6): AC expects the reviewer to remain positive.

Reviewer RmhY (4): AC expects the reviewer to increase the score as the reviewer's concerns and misunderstandings are answered.

---

### Decision · Program_Chairs · 2026-01-26

Accept (Poster)